# NEURALMARK: ADVANCING WHITE-BOX NEURAL NETWORK WATERMARKING

## ABSTRACT

As valuable digital assets, deep neural networks require ownership protection, making neural network watermarking (NNW) a promising solution. In this paper, we propose a *NeuralMark* method to advance white-box NNW, which can be seamlessly integrated into various network architectures. NeuralMark first establishes a hash mapping between the secret key and the watermark, enabling resistance to forging attacks. The watermark then functions as a filter to select model parameters for embedding, providing resilience against overwriting attacks. Furthermore, NeuralMark utilizes average pooling to defend against fine-tuning and pruning attacks. Theoretically, we analyze its security boundary. Empirically, we verify its superiority across 14 distinct Convolutional and Transformer architectures, covering five image classification tasks and one text generation task. The source codes are available at https://anonymous.4open.science/r/NeuralMark.

## 1 INTRODUCTION

The advancements in artificial intelligence have led to the development of numerous deep neural networks, particularly large language models (Mann et al., 2020; Achiam et al., 2023; Bai et al., 2023; Liu et al., 2023b; Dubey et al., 2024). Training such models requires substantial investments in human resources, computational power, and other resources, as exemplified by GPT-4, which costs around $40 million to train (Cottier et al., 2024). Thus, they can be regarded as valuable digital assets, necessitating urgent measures for ownership protection. To this end, neural network watermarking (NNW) methods (Sun et al., 2023; Lukas et al., 2022; Xue et al., 2021) have been proposed to protect model ownership by embedding watermarks within the neural network. Methods requiring access to model weights for watermark embedding and verification fall within the field of white-box NNW (Uchida et al., 2017; Liu et al., 2021; 2023a; Li et al., 2024), whereas those that do not require access to the weights belong to black-box NNW (Adi et al., 2018; Le Merrer et al., 2020; Jia et al., 2022; Li et al., 2023; He et al., 2024). Both fields have made significant progress in safeguarding model ownership. Given the distinct challenges in each field, this paper focuses on advancing white-box NNW, leaving black-box NNW for future research.

Existing white-box NNW methods can be broadly categorized into three main sub-branches: weight-based (Uchida et al., 2017; Li et al., 2021b; Liu et al., 2021; Li et al., 2024), passport-based (Fan et al., 2019; 2021; Zhang et al., 2020; Liu et al., 2023a), and activation-based (Rouhani et al., 2019; Li et al., 2021a; Lim et al., 2022) methods. Weight-based methods embed watermarks directly into model weights, offering simplicity and adaptability to various architectures. However, they are vulnerable to forging and overwriting attacks. To mitigate the vulnerabilities, passport-based methods propose binding the model performance to the watermarks by introducing sophisticated passport layers. Nevertheless, Liu et al. (2023a) argue that this binding alone is insufficient to defend against forgery and often demands an additional training time equal to the original. Activation-based methods embed watermarks in the activation maps using more complex mechanisms, yet they remain susceptible to forging attacks. Building on the distinct characteristics of those methods, we are particularly drawn to weight-based methods due to their simplicity and practicability. On one hand, unlike passport-based methods, weight-based methods do not require complex passport layers or incur additional training burdens. On the other hand, unlike activation-based methods, they do not directly constrain the activation maps for watermark embedding. However, the aforementioned limitations of existing weight-based methods motivate us to study the following question: "*How can we design a more effective and robust weight-based NNW method to address those limitations?*"

To pursue a promising solution, we propose a *NeuralMark* method, which can be seamlessly integrated into various network architectures. In the watermark generation stage, a hash mapping between the secret key and the watermark[1] is established to resist forging attacks by leveraging the avalanche effect of hash functions, where even minor changes in the input produce significantly different outputs (Liu et al., 2023a). During the watermark embedding process, the watermark functions as a filter to select model parameters for embedding. This mechanism makes it significantly more challenging for adversaries to ascertain and manipulate the filtered parameters, effectively mitigating interference with the original watermark, even when adversaries increase the embedding strength of their own watermark during overwriting attacks. To defend against fine-tuning and pruning attacks, an average pooling mechanism is applied to the filtered parameters due to its resilience against parameter perturbations. Upon obtaining the resulting parameters, we embed the watermark into those parameters using a lightweight watermarking embedding loss without compromising model performance. When a potentially unauthorized model is identified, the corresponding watermark can be extracted for ownership verification. As a result, NeuralMark demonstrates robust resistance to forging, overwriting, fine-tuning, and pruning attacks while preserving model performance.

The main contributions of this paper are three-fold. (1) To our best knowledge, there is no existing method that utilizes the watermark as a filter for selecting model parameters to resist overwriting attacks of varying strength levels. (2) We propose the NeuralMark, which, to our humble knowledge, is the first to incorporate hash mapping, watermark filtering, and average pooling mechanisms in a unified method. Also, we provide a theoretical analysis of its security boundary. (3) Experiments across 14 distinct Convolutional and Transformer architectures, covering five image classification tasks and one text generation task, verify the effectiveness and robustness of NeuralMark.

## 2 RELATED WORK

**Weight-based method**. This family of methods embeds watermarks into the model weights in neural networks (Uchida et al., 2017; Feng & Zhang, 2020; Li et al., 2021b; Liu et al., 2021). For instance, Uchida et al. (2017) propose the first weight-based method, which embeds the watermark into the model weights of an intermediate layer in the neural network. Another example is that Li et al. (2021b) propose a method based on spread transform dither modulation that enhances the secrecy of the watermark. However, those two methods cannot effectively resist forging and overwriting attacks. Moreover, Feng & Zhang (2020) utilize the secret keys to pseudo-randomly select weights for watermark embedding and apply spread-spectrum modulation to disperse the modulated watermark across different layers. This method effectively defends overwriting attacks while neglecting forging attacks. Additionally, Liu et al. (2021) propose to greedily choose important model parameters for watermark embedding without an additional secret key. Although this method is effective against forging attacks, it fails to provide strong resistance to overwriting attacks. Recently, Li et al. (2024) utilize random noises for watermark embedding and then employ a majority voting scheme to aggregate the results from multiple verification rounds. While this method improves the watermark's robustness to some extent, it is not effective in resisting forging and overwriting attacks.

**Passport-based Method**. This group of methods (Fan et al., 2019; 2021; Zhang et al., 2020; Liu et al., 2023a) integrates the watermark into the normalization layers in neural networks. Specifically, Fan et al. (2019; 2021) propose the first passport-based method, which utilizes additional passport samples (*e.g.*, images) to generate affine transformation parameters for the normalization layers, tightly binding them to the model performance. Subsequently, Zhang et al. (2020) integrate a private passport-aware branch into the normalization layers, which is trained jointly with the target model and is used solely for watermark verification. Recently, Liu et al. (2023a) argue that binding the model performance is insufficient to defend against forging attacks, and thus propose establishing a hash mapping between passport samples and watermarks.

**Activation-based Method**. This category of methods (Rouhani et al., 2019; Li et al., 2021a; Lim et al., 2022) incorporates watermarks into the activation maps of intermediate layers in neural networks. For instance, Rouhani et al. (2019) incorporate the watermark into the mean vector of activation maps generated by predetermined trigger samples. Similarly, Li et al. (2021a) directly integrate the watermark into the activation maps associated with the trigger samples. Additionally, Lim et al. (2022) embed the watermark into the hidden memory state of a recurrent neural network.

---

[1] In this paper, the watermark refers to a binary vector consisting of ones and zeros.

In summary, weight-based methods, while straightforward, often lack robustness against forging and overwriting attacks. Passport-based methods enhance robustness by binding the watermark to model performance but incur significant training overhead and remain vulnerable to overwriting attacks. Similarly, activation-based methods improve robustness by associating the watermark with activation maps, yet they lack flexibility and fail to effectively defend against forging attacks.

## 3 PROBLEM FORMULATIONS

In this section, we present several important problem formulations utilized in this paper.

### 3.1 WHITE-BOX NNW

In the white-box NNW problem, we are provided with a training dataset $\mathcal{D}$ and a white-box watermark tuple $\mathcal{W} = \{\mathbf{K}, \mathbf{b}\}$, where $\mathbf{K}$ is a secret key and $\mathbf{b}$ is a watermark. The goal is to train a watermarked model $\mathbb{M}(\boldsymbol{\theta}^*)$ using $\mathcal{D}$ such that the model parameters $\boldsymbol{\theta}^*$ effectively embed $\mathbf{b}$ while satisfying the following criteria: (i) The watermark should minimally affect the model performance and remain difficult for adversaries to detect; and (ii) The watermark must be resilient against a wide range of adversarial attacks.

### 3.2 SUCCESS CRITERIA FOR WATERMARKING ATTACKS

Building on the insights from (Fan et al., 2019; 2021; Zhu et al., 2020; Li et al., 2022), we propose that for an adversary to successfully attack a watermarked model, they must either forge a counterfeit watermark without altering the model parameters or remove the original watermark by modifying them. If the adversary only embeds a counterfeit watermark without removing the original one by modifying the model parameters, the resulting model will contain both watermarks. In this case, the model owner can submit a model containing solely the original watermark to an authoritative third-party verification agency. In contrast, the adversary cannot provide a model with only the counterfeit watermark, as they have not successfully removed the original watermark. Accordingly, the adversary cannot prove innocence unless they develop a new model embedded with only their counterfeit watermark. This not only makes stealing the original model unnecessary but also incurs significant training costs. Thus, we define three levels of success criteria for watermarking attacks. (1) **Level I**: Forging a counterfeit watermark that successfully passes the watermark verification process without modifying model parameters. (2) **Level II**: Removing the original watermark by modifying model parameters, without embedding a counterfeit one, while maintaining model performance. (3) **Level III**: Removing the original watermark and embedding a counterfeit one by modifying model parameters, while maintaining model performance.

### 3.3 THREAT MODEL

We assume that an adversary can illegally obtain a watermarked model and identify the watermarked layers. Furthermore, the adversary has access to training datasets but is constrained by limited computational resources. Based on the defined success criteria for watermarking attacks, the adversary can launch the following attacks. (1) **Forging Attack**: the adversary performs forging attacks to forge a pair of counterfeit secret key and watermark without altering the model parameters. Specifically, we employ reverse engineering attacks (Fan et al., 2019; 2021), which involve randomly forging a counterfeit watermark and subsequently deriving a corresponding secret key by freezing the model parameters. (2) **Removal + Forging Attack**: the adversary first performs removal attacks followed by forging attacks. The former aims to destroy the original watermark, while the latter attempts to forge a counterfeit watermark to pass the watermark verification process. For the removal attack, we consider widely-used fine-tuning and pruning attacks (Uchida et al., 2017; Fan et al., 2019; 2021; Liu et al., 2023a). (3) **Overwriting Attack**: the adversary removes the original watermark by embedding a counterfeit watermark (Liu et al., 2021).

## 4 METHODOLOGY

In this section, we introduce the proposed NeuralMark method.

Figure 1: Illustration of watermark filtering. Here, the model owner's watermark is $[1, 0, 1, 0]$, while the adversary's is $[0, 1, 1, 0]$. Without filtering, all 16 parameters overlap. After one round of filtering, each retains eight parameters, with four overlapping. A second round leaves four parameters each, with no overlap.

### 4.1 MOTIVATION

As stated above, there are three types of attacks: (i) forging attack, (ii) removal + forging attack; and (iii) overwriting attack. These attacks motivate the development of Neuralmark.

To counter forging attacks, we draw inspiration from Liu et al. (2023a), which builds a hash mapping between passport samples and the watermark to resist such attacks. However, it necessitates replacing normalization layer parameters (*e.g.*, batch normalization) with those generated from passport samples for the same purpose, which complicates practical deployment and is unnecessary. To address those issues, we propose to directly establish a hash mapping between the secret key and the watermark, which is simple and practical. Any attempt to learn the secret key and watermark would require breaking the underlying cryptographic hash function, which is computationally infeasible due to its avalanche effect, where even small changes in the input result in significantly different outputs (Liu et al., 2023a). To resist removal attacks, we utilize the widely-used average pooling mechanism (Uchida et al., 2017; Liu et al., 2021), which aggregates parameters across broader regions, enhancing robustness against parameter perturbations caused by fine-tuning or pruning attacks.

To defend against overwriting attacks, the watermarked parameters need to be as secret as possible. The model owner's watermark is private and consists of a binary vector with randomly arranged ones and zeros, providing a promising solution: *Utilizing it as a private filter for model parameters*. Since the watermarks of the adversary and the model owner are distinct, the overlap in the model parameters after filtering will be reduced. As exemplified in Figure 1, the model owner's watermark is $[1, 0, 1, 0]$, while the adversary's is $[0, 1, 1, 0]$. Without filtering, all 16 model parameters overlap, resulting in a $100\%$ overlap ratio. After one round of filtering, each party obtains eight parameters, with four overlapping, leading to a $50\%$ overlap ratio. Following a second round of filtering, each party has four parameters, with no overlap and a $0\%$ overlap ratio. This illustrates that as filtering progresses, the parameter overlap between the model owner and the adversary effectively decreases. Hence, embedding the watermark into the filtered parameters can mitigate the overwriting attack.

In summary, those mechanisms are fundamental to NeuralMark. Next, we elaborate on NeuralMark.

### 4.2 NEURALMARK

NeuralMark includes three primary steps: (i) watermark generation; (ii) watermark embedding; and (iii) watermark verification.

**Watermark Generation**. As aforementioned, we establish a hash mapping between the secret key and the watermark. Specifically, the watermark $\mathbf{b}$ is generated as $\mathbf{b} = \mathcal{H}(\mathbf{K}) \in \{0, 1\}^n$, where each element in $\mathbf{K} \in \mathbb{R}^{k \times n}$ is drawn from a random distribution (*e.g.*, Gaussian distributions), $\mathcal{H}(\cdot)$ denotes a hash function (*e.g.*, SHAKE-256 (Dworkin, 2015)), and $n$ represents the length of watermark. As a result, hash mapping effectively defends against forging attacks. Furthermore, in several practical scenarios where a model is collaboratively developed by multiple owners, their signatures can be seamlessly integrated into NeuralMark to facilitate ownership verification. Due to the page limit, additional discussion is provided in Appendix B.1.

**Watermark Embedding**. We now introduce the step-by-step process for embedding the watermark $\mathbf{b}$ into the model $\mathbb{M}(\boldsymbol{\theta})$. As illustrated in Figure 2(a), we first randomly select and flatten a subset of parameters from $\boldsymbol{\theta}$ into a parameter vector $\mathbf{w} \in \mathbb{R}^m$. Then, we perform the following operations:

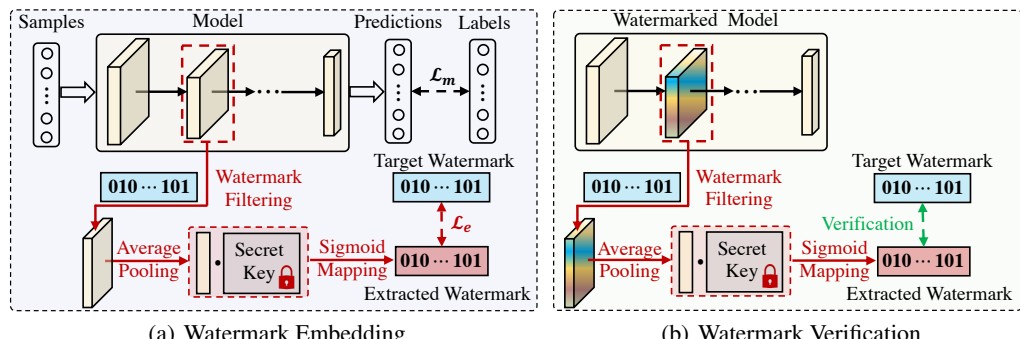

(a) Watermark Embedding  (b) Watermark Verification

Figure 2: Illustrations of the processes for watermark embedding (a) and verification (b).

- *Watermark Filtering*: Let $\mathbf{w}^{(0)} = \mathbf{w}$ be the initial parameter vector. In the $r$-th ($r \in \{1, \cdots, R\}$) filtering round, the watermark $\mathbf{b}$ is repeated to match the length of $\mathbf{w}^{(r-1)}$, forming $\mathbf{b}^{(r)}$, with any excess parameters in $\mathbf{w}^{(r-1)}$ discarded. Subsequently, the parameter vector $\mathbf{w}^{(r)}$ is constructed by selecting the elements from $\mathbf{w}^{(r-1)}$ at positions where $\mathbf{b}^{(r)}$ equals one, *i.e.*, $\mathbf{w}^{(r)} = \big[ w_i^{(r-1)} \mid i \in \{j \mid b_j^{(r)} = 1\} \big]$, where $w_i^{(r-1)}$ is the $i$-th element of $\mathbf{w}^{(r-1)}$, and $b_j^{(r)}$ is the $j$-th element of $\mathbf{b}^{(r)}$.

- *Average Pooling*: After completing watermark filtering, we obtain the final parameter vector $\mathbf{w}^{(R)}$. Next, based on the first dimension $k$ of $\mathbf{K}$, we reshape $\mathbf{w}^{(R)}$ into a matrix $\mathbf{W}$ with dimensions $-1 \times k$, where $-1$ is automatically inferred from the length of $\mathbf{w}^{(R)}$, and any remaining parameters that do not fit are discarded. Finally, we perform average pooling along the first dimension of matrix $\mathbf{W}$ to obtain the final parameter vector $\widetilde{\mathbf{w}}$.

- *Sigmoid Mapping*: Building on $\widetilde{\mathbf{w}}$ and $\mathbf{K}$, we utilize the sigmoid function $\delta(\cdot)$ to calculate the extracted watermark $\widetilde{\mathbf{b}}$, *i.e.*, $\widetilde{\mathbf{b}} = \delta(\widetilde{\mathbf{w}}\mathbf{K})$.

- *Objective Optimization*: We formulate the watermark embedding loss $\mathcal{L}_e$ as

$$\mathcal{L}_e = -\frac{1}{n} \sum_{i=1}^{n} \big[ b_i \ln(\widetilde{b}_i) + (1 - b_i) \ln(1 - \widetilde{b}_i) \big], \tag{1}$$

where $b_i$ and $\widetilde{b}_i$ are $i$-th elements of $\mathbf{b}$ and $\widetilde{\mathbf{b}}$, respectively. To minimize the impact of watermark embedding on the model performance, we jointly optimize this task alongside the main task. Thus, the final optimization objective is formulated as

$$\min_{\theta} \mathcal{L}_m + \lambda \mathcal{L}_e, \tag{2}$$

where $\mathcal{L}_m$ denotes the main task loss (*e.g.*, classification loss), and $\lambda$ is a positive trade-off hyperparameter. By minimizing Eq. (2), the watermark can be embedded into model parameters during the main task training. The embedding process is summarized in Algorithm 1 within Appendix A.

**Watermark Verification**. The watermark verification process is similar to the embedding process, as depicted in Figure 2(b). Concretely, upon identifying a potentially unauthorized model, the relevant subset of model parameters is extracted and subjected to watermark filtering, average pooling, and sigmoid mapping to derive an extracted watermark $\widetilde{\mathbf{b}}$. This extracted watermark $\widetilde{\mathbf{b}}$ is then compared to the model owner's watermark $\mathbf{b}$ using the watermark detection rate, which is defined by

$$\rho = \frac{1}{n} \sum_{i=1}^{n} \mathbb{1} \big[ b_i, \mathcal{T}(\widetilde{b}_i) \big], \tag{3}$$

where $\mathcal{T}(x)$ is a threshold function that assigns a value of one for $x > 0.5$ and zero for $x \leq 0.5$, and $\mathbb{1}(\psi)$ is an indicator function that evaluates to one if $\psi$ is true and to zero otherwise. The unauthorized model is confirmed to belong to the model owner if the following conditions are satisfied: (i) The watermark detection rate $\rho$ exceeds a theoretical security boundary $\rho^*$, which will be analyzed later; and (ii) The watermark must correspond to the output of the hash function applied to the secret key, ensuring cryptographic consistency with the predefined hash mapping (please refer to Appendix B.2 for a detailed analysis). The verification process is outlined in Algorithm 2 within Appendix A.

### 4.3 THEORETICAL ANALYSIS

We present a theoretical analysis to determine the security boundary in Theorem 1.

**Theorem 1** *Under the assumption that the hash function produces uniformly distributed outputs (Bellare & Rogaway, 1993), for a model watermarked by NeuralMark with a watermark tuple $\{\mathbf{K}, \mathbf{b}\}$, where $\mathbf{b} = \mathcal{H}(\mathbf{K})$, if an adversary attempts to forge a counterfeit watermark tuple $\{\mathbf{K}', \mathbf{b}'\}$ such that $\mathbf{b}' = \mathcal{H}(\mathbf{K}')$ and $\mathbf{K}' \neq \mathbf{K}$, then the probability of achieving a watermark detection rate of at least $\rho$ (i.e., $\geq \rho$) is upper-bounded by $\frac{1}{2^n} \sum_{i=0}^{n-\lceil \rho n \rceil} \binom{n}{i}$.*

The proof of Theorem 1 is provided in Appendix C. Theorem 1 provides a theoretical benchmark for establishing the security boundary of the watermark detection rate. Specifically, with $n = 256$, if the watermark detection rate $\rho \geq 88.28\%$, the probability of this occurring by forgery is less than $1/2^{128}$. This negligible probability allows us to confirm ownership with high confidence. Thus, we set $n = 256$ and use $88.28\%$ as the security bound for the watermark detection rate in the experiments.

### 4.4 COMPARISON WITH RELATED STUDIES

We now compare NeuralMark with several existing studies. To our humble knowledge, the most closely related watermarking methods are presented in (Uchida et al., 2017), (Liu et al., 2021), and (Li et al., 2024), referred to as VanillaMark, GreedyMark, and VoteMark, respectively. VanillaMark serves as the foundation for GreedyMark, VoteMark, and NeuralMark. However, VanillaMark and VoteMark are ineffective in defending against forging and overwriting attacks, while GreedyMark does not effectively resist overwriting attacks. More comparison details are offered in Appendix B.3.

## 5 EXPERIMENTS

In this section, we evaluate NeuralMark across a variety of datasets, architectures, and tasks.

### 5.1 EXPERIMENTAL SETUP

**Datasets and Architectures**. We adopt five image classification datasets: CIFAR-10, CIFAR-100 (Krizhevsky et al., 2009), Caltech-101 (Fei-Fei et al., 2004), Caltech-256 (Griffin et al., 2007), and TinyImageNet (Le & Yang, 2015), as well as one text generation dataset, E2E (Novikova et al., 2017). Also, we utilize 12 image classification architectures, including eight Convolutional architectures: AlexNet (Krizhevsky et al., 2012), VGG-13, VGG-16 (Simonyan & Zisserman, 2015), GoogLeNet (Szegedy et al., 2015), ResNet-18, ResNet-34 (He et al., 2016), WideResNet-50 (Zagoruyko, 2016), and MobileNet-V3-L (Howard et al., 2019), as well as four Transformer architectures: ViT-B/16, ViT-B/32 (Dosovitskiy, 2021), Swin-V2-B, and Swin-V2-S (Liu et al., 2022). Furthermore, we employ two text generation architectures: GPT-2-S and GPT-2-M (Radford et al., 2019).

**Baselines and Metrics**. We compare NeuralMark with VanillaMark (Uchida et al., 2017), Greedy-Mark (Liu et al., 2021), and VoteMark (Li et al., 2024). Additionally, we include a comparison with a method that does not involve watermark embedding, referred to as Clean. For the image classification task, we assess model performance using classification accuracy, while the watermark embedding task is evaluated based on the watermark detection rate. Following the methodology of (Hu et al., 2022), we evaluate the text generation task using BLEU, NIST, MET, ROUGE-L, and CIDEr metrics. More experimental details are provided in Appendix D.

Table 1: Comparison of classification accuracy (%) across distinct datasets using AlexNet and ResNet-18, respectively. Watermark detection rates are omitted as they all reach 100%.

| Dataset | Clean | | NeuralMark | | VanillaMark | | GreedyMark | | VoteMark | |
|---|---|---|---|---|---|---|---|---|---|---|
| | AlexNet | ResNet-18 | AlexNet | ResNet-18 | AlexNet | ResNet-18 | AlexNet | ResNet-18 | AlexNet | ResNet-18 |
| CIFAR-10 | 91.05 | 94.76 | 90.93 | 94.50 | 91.01 | 94.87 | 90.88 | 94.69 | 90.86 | 94.79 |
| CIFAR-100 | 68.24 | 76.23 | 68.57 | 76.34 | 68.43 | 76.22 | 68.31 | 76.14 | 68.53 | 76.74 |
| Caltech-101 | 68.07 | 68.83 | 68.38 | 68.47 | 68.54 | 68.99 | 68.59 | 69.08 | 68.88 | 67.91 |
| Caltech-256 | 44.27 | 54.09 | 44.55 | 53.71 | 44.73 | 53.47 | 44.64 | 53.28 | 44.43 | 54.71 |
| TinyImageNet | 42.42 | 53.48 | 42.31 | 53.22 | 42.50 | 53.36 | 42.94 | 53.31 | 42.50 | 53.47 |

## 5.2 FIDELITY EVALUATION

First, we evaluate the influence of watermark embedding on the model performance across diverse datasets. Table 1 reports the results across five image datasets using AlexNet and ResNet-18 architectures. We observe that all methods have minimal impact on model performance while successfully embedding watermarks, indicating that NeuralMark and other methods maintain model performance across diverse datasets during watermark embedding. We then assess the impact of NeuralMark on model performance across various architectures. Table 2 lists the results of NeuralMark on the CIFAR-100 dataset using VGG-13, VGG-16, GoogLeNet, ResNet-34, WideResNet-50, MobileNet-V3-L, ViT-B/16, ViT-B/32, Swin-V2-B, and Swin-V2-S architectures. We can see that NeuralMark maintains a $100\%$ watermark detection rate across a wide range of architectures while exerting minimal impact on model performance. Those observations suggest that NeuralMark exhibits generalizability across diverse architectures. Finally, we evaluate the impact of NeuralMark on the performance of text generation tasks. Table 3 presents the results of NeuralMark applied to the GPT-2-S and GPT-2-M architectures on the E2E dataset. We can observe that NeuralMark achieves a $100\%$ watermark detection rate while maintaining nearly lossless model performance. Those results validate the potential of NeuralMark in safeguarding the ownership of generative models. To summarize, NeuralMark demonstrates consistent fidelity across various datasets, architectures, and tasks.

Table 2: Comparison of classification accuracy (%) on the CIFAR-100 dataset using various architectures. Watermark detection rates are omitted as they all reach 100%.

| Method | ViT-B/16 | ViT-B/32 | Swin-V2-B | Swin-V2-S | VGG-16 | VGG-13 | ResNet-34 | WideResNet-50 | GoogLeNet | MobileNet-V3-L |
|---|---|---|---|---|---|---|---|---|---|---|
| Clean | 39.07 | 29.94 | 52.99 | 55.88 | 72.75 | 72.71 | 77.06 | 59.67 | 60.71 | 61.11 |
| NeuralMark | 39.22 | 29.13 | 53.57 | 55.87 | 72.61 | 71.49 | 77.03 | 58.41 | 60.02 | 61.8 |

Table 3: Comparison on the E2E dataset using GPT-2-S and GPT-2-M, respectively. Watermark detection rates are omitted as they all reach 100%.

| GPT-2-S | BLEU | NIST | MET | ROUGE-L | CIDEr | GPT-2-M | BLEU | NIST | MET | ROUGE-L | CIDEr |
|---|---|---|---|---|---|---|---|---|---|---|---|
| Clean | 69.36 | 8.76 | 46.06 | 70.85 | 2.48 | Clean | 68.7 | 8.69 | 46.38 | 71.19 | 2.5 |
| NeuralMark | 69.59 | 8.79 | 46.01 | 70.85 | 2.48 | NeuralMark | 67.73 | 8.57 | 46.07 | 70.66 | 2.47 |

## 5.3 ROBUSTNESS EVALUATION

**Forging Attack**. We adopt the setting detailed in Section 3.3 to assess the robustness of NeuralMark against forging attacks. Concretely, for VanillaMark and VoteMark, we first randomly generate a counterfeit watermark and then learn the corresponding secret key by freezing the model parameters. Since GreedyMark does not

Table 4: Comparison of resistance to forging attacks using ResNet-18.

| Dataset | NeuralMark | VanillaMark | GreedyMark | VoteMark |
|---|---|---|---|---|
| CIFAR-10 | 48.56 | 100.00 | 50.70 | 100.00 |
| CIFAR-100 | 49.41 | 100.00 | 52.85 | 100.00 |

require a secret key, we utilize 10 sets of randomly forged watermarks to directly verify them using the watermarked model. For NeuralMark, due to the avalanche effect of hash functions, a method similar to GreedyMark is employed, where 10 sets of randomly forged watermarks are directly verified using the watermarked model. Table 4 presents the watermark detection rates of forging attacks, we present the following significant observations. (1) For VanillaMark and VoteMark, a pair of counterfeited secret key and watermark can be successfully learned through reverse engineering, as they are not specifically designed to withstand forging attacks. (2) NeuralMark and GreedyMark demonstrate robust resistance against forging attacks, which aligns with our expectations.

**Removal + Forging Attack**. We adhere to the setting stated in Section 3.3 to evaluate the robustness of NeuralMark against removal + forging attacks.

First, we conduct fine-tuning attacks followed by forging attacks. Following Liu et al. (2021), for all fine-tuning attacks, we use the same hyper-parameters as during training, except for setting the learning rate to 0.001. Then, we replace the task-specific classifier and minimize the main task loss $\mathcal{L}_m$ to optimize all parameters for 100 epochs. Table 5 reports the results of fine-tuning attacks, we can make several meaningful observations. (1) Watermarks embedded with NeuralMark maintain a $100\%$ watermark detection rate across all fine-tuning tasks. In contrast, watermarks embedded with VanillaMark, GreedyMark, and VoteMark experience a slight reduction in detection rates across

Table 5: Comparison of resistance to fine-tuning attacks using ResNet-18. Values (%) inside and outside the bracket are watermark detection rate and classification accuracy, respectively.

| Fine-tuning | Clean | | NeuralMark | | VanillaMark | | GreedyMark | | VoteMark | |
|---|---|---|---|---|---|---|---|---|---|---|
| | AlexNet | ResNet-18 | AlexNet | ResNet-18 | AlexNet | ResNet-18 | AlexNet | ResNet-18 | AlexNet | ResNet-18 |
| CIFAR-100 to CIFAR-10 | 89.44 | 93.21 | 89.11(100) | 93.74(100) | 89.00(100) | 93.29(100) | 89.34(99.21) | 93.21(100) | 89.03(100) | 93.59(100) |
| CIFAR-10 to CIFAR-100 | 65.46 | 72.17 | 64.60(100) | 71.67(100) | 65.03(92.18) | 72.49(97.26) | 64.57(98.82) | 72.06(100) | 64.83(96.09) | 72.27(98.04) |
| Caltech-256 to Caltech-101 | 72.69 | 76.93 | 73.55(100) | 76.60(100) | 72.90(100) | 78.48(100) | 73.12(100) | 77.19(100) | 72.90(100) | 77.41(100) |
| Caltech-101 to Caltech-256 | 43.39 | 46.48 | 43.15(100) | 44.42(100) | 43.21(98.43) | 45.69(99.60) | 43.47(99.60) | 45.25(100) | 43.78(98.43) | 45.29(100) |

several tasks. Those results indicate that fine-tuning attacks cannot effectively remove watermarks embedded with NeuralMark. (2) All methods exhibit similar model performance after fine-tuning. This implies that NeuralMark and other methods do not significantly impact model performance after fine-tuning. Furthermore, Table 9 in Appendix E.1 reports the experimental results of fine-tuning the watermark embedding layer and classifier. As can be seen, the watermark detection rate remains at 100%, but the model performance of all methods exhibits a substantial decline. Specifically, for the CIFAR-10 to CIFAR-100 task using ResNet-18, the accuracy achieved by NeuralMark through fine-tuning the watermark embedding layer and classifier is 49.77%, which is markedly lower than the 71.67% accuracy obtained when all parameters are fine-tuned. Those results indicate that solely fine-tuning the watermark embedding layer and classifier makes it challenging to ensure effective model performance. Consequently, we do not consider this type of fine-tuning attack in the subsequent experiments. After conducting fine-tuning attacks, we perform forging attacks adhering to the same settings detailed in **Forging Attack**. From Table 6, we observe a phenomenon similar to that in Table 4, which further demonstrates that NeuralMark effectively resists forging attacks.

Table 6: Comparison of resistance to forging attacks after fine-tuning attacks and pruning attacks (with a pruning ratio of 40%) using ResNet-18.

| Dataset | NeuralMark | | VanillaMark | | GreedyMark | | VoteMark | |
|---|---|---|---|---|---|---|---|---|
| | Fine-tuning + Forging | Pruning + Forging | Fine-tuning + Forging | Pruning + Forging | Fine-tuning + Forging | Pruning + Forging | Fine-tuning + Forging | Pruning + Forging |
| CIFAR-10 | 48.90 | 49.14 | 100.00 | 100.00 | 49.30 | 49.30 | 100.00 | 100.00 |
| CIFAR-100 | 48.82 | 49.37 | 100.00 | 100.00 | 49.30 | 50.27 | 100.00 | 100.00 |

Then, we perform pruning attacks followed by forging attacks. In pruning attacks, we randomly reset a specified proportion of model parameters in the watermark embedding layer to zero. Figure 3 shows the results of pruning attacks on the CIFAR-100 dataset. We can observe that as the pruning ratio increases, the performance of NeuralMark degrades while the detection rate remains nearly 100%. This indicates NeuralMark's robustness against pruning attacks, primarily due to the average pooling mechanism, which mitigates the effects of parameter pruning by aggregating parameters across broader regions. Moreover, we observe that both VanillaMark and VoteMark exhibit strong resistance to pruning attacks, while GreedyMark demonstrates relatively weak resistance. One possible reason is that GreedyMark depends on several important parameters, and their removal may affect its robustness. More experimental results of pruning attacks across distinct datasets are provided in Appendix E.2. Following pruning attacks, we conduct forging attacks following the same settings stated in **Forging Attack**. Table 6 presents the results of forging attacks at a pruning ratio of 40%, we can see that NeuralMark remains robust against forging attacks, even with 40% of parameters pruned. Moreover, Table 10 in Appendix E.2 lists more forging attack results with various pruning ratios. As can be seen, NeuralMark can effectively resist forging attacks in all scenarios.

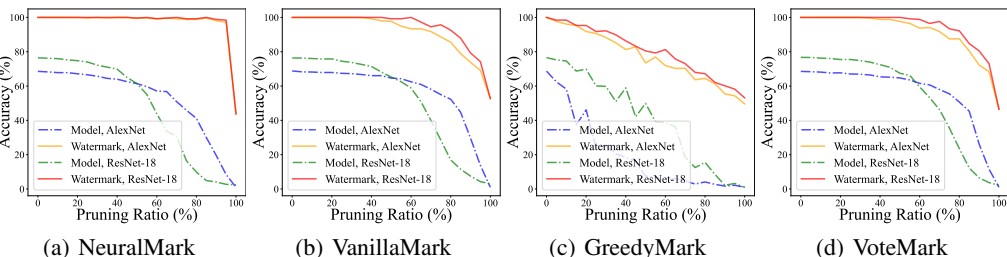

(a) NeuralMark  (b) VanillaMark  (c) GreedyMark  (d) VoteMark

Figure 3: Comparison of resistance to pruning attacks at various pruning ratios on the CIFAR-100 dataset using AlexNet and ResNet-18, respectively.

Overall, the removal + forging attack cannot remove watermarks embedded using NeuralMark, nor can it forge watermarks that satisfy NeuralMark's criteria.

Table 7: Comparison of resistance to overwriting attacks at various trade-off hyper-parameters ($\lambda$) and learning rates ($\eta$). Values (%) inside and outside the bracket are watermark detection rate and classification accuracy, respectively.

| Overwriting | $\lambda$ | NeuralMark | VanillaMark | GreedyMark | VoteMark | $\eta$ | NeuralMark | VanillaMark | GreedyMark | VoteMark |
|---|---|---|---|---|---|---|---|---|---|---|
| CIFAR-100 to CIFAR-10 | 1 | 93.65 (100) | 93.30 (100) | 93.45 (48.82) | 93.63 (100) | 0.001 | 93.65 (100) | 93.30 (100) | 93.45 (48.82) | 93.63 (100) |
| | 10 | 93.44 (100) | 93.58 (100) | 93.29 (51.17) | 93.13 (100) | 0.005 | 91.76 (99.60) | 92.17 (73.04) | 92.13 (50.00) | 92.45 (78.90) |
| | 50 | 93.46 (100) | 93.50 (100) | 93.07 (55.07) | 93.39 (100) | 0.01 | 91.58 (92.18) | 91.79 (62.10) | 91.53 (49.60) | 91.76 (60.15) |
| | 100 | 93.53 (100) | 92.95 (94.53) | 93.18 (54.29) | 93.53 (96.48) | 0.1 | 75.2 (50.78) | 79.68 (47.26) | 72.42 (53.12) | 70.92 (54.29) |
| | 1000 | 93.09 (100) | 92.89 (53.90) | 92.85 (49.60) | 92.77 (59.37) | 1 | 10.00 (44.53) | 10.00 (53.51) | 10.00 (48.04) | 10.00 (53.51) |
| CIFAR-10 to CIFAR-100 | 1 | 71.78 (100) | 72.68 (98.82) | 71.34 (55.07) | 72.97 (98.43) | 0.001 | 71.78 (100) | 72.68 (98.82) | 71.34 (55.07) | 72.97 (98.43) |
| | 10 | 72.6 (100) | 72.03 (98.04) | 72.30 (49.21) | 72.08 (98.04) | 0.005 | 71.04 (99.60) | 70.02 (69.53) | 70.25 (48.04) | 71.11 (71.09) |
| | 50 | 72.73 (100) | 72.45 (95.70) | 70.92 (46.87) | 72.38 (97.26) | 0.01 | 69.14 (96.48) | 69.02 (59.76) | 69.25 (46.09) | 68.88 (62.11) |
| | 100 | 71.49 (100) | 71.92 (92.18) | 72.05 (48.04) | 72.72 (93.75) | 0.1 | 51.88 (60.54) | 51.76 (53.90) | 51.71 (51.56) | 51.74 (56.25) |
| | 1000 | 71.81 (100) | 71.35 (57.42) | 71.74 (51.95) | 70.73 (56.64) | 1 | 1.00 (44.53) | 1.00 (53.15) | 1.00 (50.00) | 1.00 (53.51) |

**Overwriting Attack**. We follow the setting outlined in Section 3.3 to assess the robustness of NeuralMark against overwriting attacks. We analyze two key factors: the hyper-parameter $\lambda$ in Eq. (2) and the learning rate $\eta$. Here, $\lambda$ controls the strength of the watermark embedding, with larger values leading to stronger embedding, while $\eta$ primarily affects model performance.

First, we investigate the influence of $\lambda$ in overwriting attacks. Specifically, we set $\lambda$ to 1, 10, 50, 100, and 1000, respectively. Table 7 presents the results on the CIFAR-100 to CIFAR-10 and CIFAR-10 to CIFAR-100 tasks using ResNet-18. We report only the original watermark detection rate, as the adversary's watermark detection rate reaches $100\%$. Also, as defined in the success criterion Level III in Section 3.2, the original watermark must be effectively removed for overwriting attacks to be deemed successful. Thus, the overwriting attack experiments focus solely on whether the original watermark can be successfully removed. We can summarize several insightful observations. (1) As $\lambda$ increases, the original watermark detection rate of NeuralMark remains at 100%, while those of VanillaMark, GreedyMark, and VoteMark significantly decline. In particular, when $\lambda = 1000$, the embedding strength of the adversary's watermark is 1000 times greater than that of the original watermark. At this point, the original watermark detection rates for NeuralMark, VanillaMark, GreedyMark, and VoteMark on the CIFAR-100 to CIFAR-10 tasks are 100%, 53.90%, 49.60%, and 59.37%, respectively. Those results indicate that NeuralMark exhibits strong robustness against overwriting attacks, primarily due to the watermark filtering mechanism, making it difficult to remove the original watermark. (2) As $\lambda$ increases, model performance remains relatively stable. This is because overwriting attacks jointly train both the main task and the watermark embedding task, enabling the model parameters to effectively adapt to both.

Then, we examine the impact of $\eta$ in overwriting attacks. Concretely, we set $\eta$ to 0.001, 0.005, 0.01, 0.1, and 1, respectively. Table 7 lists the results on the CIFAR-100 to CIFAR-10 and CIFAR-10 to CIFAR-100 tasks using ResNet-18. The observations are as follows. (1) As $\eta$ increases, model performance declines due to its substantial impact on performance. Thus, the adversary cannot arbitrarily increase $\eta$ to strengthen the attack. (2) At $\eta = 0.005$, the original watermark detection rates for VanillaMark, GreedyMark, and VoteMark drop dramatically, whereas NeuralMark maintains a detection rate close to 100%. When $\eta = 0.01$, the model performance of NeuralMark on the CIFAR-100 to CIFAR-10 task decreases by 2.07%, but its original watermark detection rate remains above the security boundary of 88.28%, while those for the other methods fall significantly. For $\eta >= 0.1$, although the original watermark detection rate of NeuralMark drops below the security boundary, the model performance is completely compromised, indicating that the attack is ineffective.

On the whole, all results confirm NeuralMark's robustness against overwriting attacks.

## 5.4 ADDITIONAL ANALYSIS

**Parameter Distribution**. To assess the secrecy of NeuralMark, Figures 4(a) and 4(b) present the parameter distributions on the CIFAR-100 dataset with ResNet-18 and ViT-B/16 architectures. As can be seen, the parameter distributions of Clean and NeuralMark are nearly indistinguishable. Thus, it is challenging for adversaries to detect the embedded watermarks within the model. More parameter distribution results are provided in Appendix E.4.

**Performance Convergence**. To examine the impact of NeuralMark on model performance convergence, Figures 4(c) and 4(d) show the results on the CIFAR-100 dataset with ResNet-18 and ViT-B/16 architectures. We find that the performance curves of Clean and NeuralMark exhibit a similar trend of

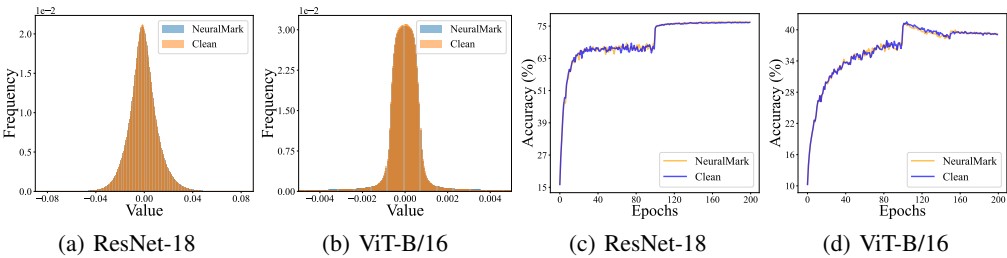

| (a) ResNet-18 | (b) ViT-B/16 | (c) ResNet-18 | (d) ViT-B/16 |

Figure 4: Comparison of parameter distributions (a, b) and performance convergences (c, d) on the CIFAR-100 dataset using distinct architectures.

change and are closely aligned, indicating that NeuralMark does not negatively affect the convergence of model performance. More performance convergence results are offered in Appendix E.5.

**Average Pooling**. To verify the efficacy of average pooling, we compare NeuralMark with its variant without average pooling, *i.e*, NeuralMark w/o AP. As shown in Table 8, both versions resist fine-tuning attacks at lower learning rates. However, at a learning rate of $0.01$, the detection rate for NeuralMark (w/o AP) drops to $81.64\%$, below the security boundary, while NeuralMark maintains at $96.87\%$. In addition, the detection rate of NeuralMark (w/o AP) rapidly declines with increasing pruning rates, reaching $69.92\%$ at an $80\%$ pruning rate, while NeuralMark achieves $99.21\%$. Those results confirm that average pooling enhances resistance to both fine-tuning and pruning attacks.

Table 8: Comparison of the effects of average pooling on resistance to fine-tuning and pruning attacks using ResNet-18. Values (%) inside and outside the bracket are watermark detection rate and classification accuracy, respectively.

| Method | CIFAR-100 to CIFAR-10 Fine-tuning Attack | | | CIFAR-100 Pruning Attack | | |
|---|---|---|---|---|---|---|
| | Learning Rate | | | Pruning Ratio | | |
| | 0.001 | 0.005 | 0.01 | 40% | 60% | 80% |
| NeuralMark (w/o AP) | 93.26 (100) | 92.20 (100) | 90.68 (81.64) | 71.82 (90.62) | 57.50 (78.51) | 16.14 (69.92) |
| NeuralMark | 93.74 (100) | 92.25 (100) | 91.25 (96.87) | 69.86 (100) | 43.88 (99.21) | 9.85 (99.21) |

**Filtering Rounds**. To analyze watermark filtering efficacy, we generate five counterfeit watermarks and compute the overlap ratio between parameters filtered with those and the original watermark. As illustrated in Figure 5, the overlap rate decreases towards zero with more filtering rounds, indicating that watermark filtering enhances the secrecy of the watermarked parameters. Furthermore, additional experiments are conducted using 6 and 8 filters to evaluate robustness against various attacks, compared to NeuralMark's default setting of 4 filters. The results are offered in Appendix E.6, indicating that NeuralMark maintains high robustness across all scenarios.

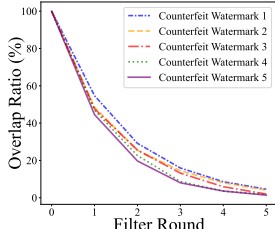

Figure 5: Comparison of parameter overlap ratio with different filter rounds on the CIFAR-100 dataset using ResNet-18.

**Additional Analyses**. Due to the page limit, we include additional analysis experiments in Appendices E.7-E.9. These include the impact of watermark embedding layers and length on model performance, along with an efficiency analysis of NeuralMark. The results demonstrate its effectiveness and efficiency.

# 6 CONCLUSION

In this paper, we present the NeuralMark, which integrates three core mechanisms: hash mapping, watermark filtering, and average pooling. The first binds secret keys to watermarks, resisting forging attacks. The second ensures the secrecy of watermarked parameters, protecting against overwriting attacks. The third enhances robustness against parameter perturbations, defending against fine-tuning and pruning attacks. Also, we provide a theoretical analysis of NeuralMark's security boundary. Extensive experiments on various datasets, architectures, and tasks confirm NeuralMark's effectiveness and robustness. We expect NeuralMark to serve as a benchmark for advancing white-box NNW. As a future direction, we plan to extend NeuralMark to more complex scenarios, for instance, federated learning (Yang et al., 2019).

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

We provide additional details and results in the appendices. Below are the contents.

- Appendix A: Algorithms for the watermark embedding and verification in NeuralMark.
- Appendix B: More detailed discussions are provided, including a discussion on watermark generation (Appendix B.1), an analysis of resisting forging attacks (Appendix B.2), comparisons with related studies (Appendix B.3), as well as the limitations and broader impact of NeuralMark (Appendix B.4).
- Appendix C: Proof of Theorem 1.
- Appendix D: Implementation details of NeuralMark.
- Appendix E: Additional experimental results.

## A    ALGORITHM OF NEURALMARK

Algorithms 1-2 offer the watermark embedding and verification processes in NeuralMark, respectively.

---

**Algorithm 1** Watermark Embedding in NeuralMark

---

**Input:** Training dataset $\mathcal{D}$, secret key $\mathbf{K}$, random index $\mathbf{I}$, and hyper-parameters $\lambda$, $T$, and $R$.
**Output:** Watermarked model $\mathbb{M}(\theta^*)$.
 1: Randomly initialize the model parameter $\theta$.
 2: Generate the watermark $\mathbf{b} = \mathcal{H}(\mathbf{K})$.
 3: **for** $t = 0$ to $T - 1$ **do**
 4:    Use $\mathbf{I}$ to select a subset from $\theta$ and flatten it to create $\mathbf{w}$.
 5:    **for** $r = 1$ to $R$ **do**
 6:       Perform watermark filtering on $\mathbf{w}$ to obtain $\mathbf{w}^{(r)}$.
 7:    **end for**
 8:    Apply average pooling on $\mathbf{w}^{(R)}$ to yield $\widetilde{\mathbf{w}}$.
 9:    Execute sigmoid mapping on $\widetilde{\mathbf{w}}\mathbf{K}$ to produce $\widetilde{\mathbf{b}}$.
10:    Update $\theta$ based on Eq. (2).
11: **end for**

---

**Algorithm 2** Watermark Verification in NeuralMark

---

**Input:** Watermarked model $\mathbb{M}(\theta^*)$, secret key $\mathbf{K}$, watermark $\mathbf{b}$, random index $\mathbf{I}$, filter rounds $R$, and security boundary $\rho^*$.
**Output:** True (Verification Success) or False (Verification Failure).
 1: Use $\mathbf{I}$ to select a subset from $\theta^*$ and flatten it to create $\mathbf{w}$.
 2: **for** $r = 1$ to $R$ **do**
 3:    Perform watermark filtering on $\mathbf{w}$ to obtain $\mathbf{w}^{(r)}$.
 4: **end for**
 5: Apply average pooling on $\mathbf{w}^{(R)}$ to yield $\widetilde{\mathbf{w}}$.
 6: Execute sigmoid mapping on $\widetilde{\mathbf{w}}\mathbf{K}$ to produce $\widetilde{\mathbf{b}}$.
 7: Calculate watermark detection rate $\rho$ based on Eq. (3).
 8: **if** $\rho \geq \rho^*$ **and** $\mathcal{H}(\mathbf{K}) = \mathbf{b}$ **then**
 9:    **return** True
10: **else**
11:    **return** False
12: **end if**

---

## B    MORE DETAILED DISCUSSIONS

### B.1    DISCUSSION ON WATERMARK GENERATION

In several practical scenarios where a model is collaboratively developed by multiple owners, their signatures can be seamlessly integrated into NeuralMark to facilitate ownership verification. Specifically, the signatures of model owners are concatenated with the secret key and then hashed to generate the watermark, *i.e.*, $\mathbf{b} = \mathcal{H}(\mathbf{S}_1||\cdots||\mathbf{S}_n||\mathbf{K}) \in \{0, 1\}^n$, where $||$ denotes concatenation operation,

and $\mathbf{S}_n$ represents the $n$-th model owner's signature, serving as cryptographic proof of its identity. Accordingly, this mechanism enables repeated public verification by multiple owners. Furthermore, its robustness in resisting forging attacks is guaranteed by the cryptographic properties of the hash function, similar to the case where $\mathbf{b} = \mathcal{H}(\mathbf{K})$. Also, this mechanism is orthogonal to NeuralMark's existing mechanisms (*i.e.*, watermark filtering and average pooling) and does not compromise its robustness against other types of attacks.

## B.2 ANALYSIS OF RESISTING FORGING ATTACKS

In this section, we analyze why the hash mapping between the secret key and watermarks can effectively resist forging attacks. On the one hand, if an adversary attempts to forge a pair of counterfeit secret key and watermark through reverse engineering while considering the hash mapping relationship, it is computationally infeasible due to the *avalanche effect* of hash functions, where even small changes in the input result in significantly different outputs (Liu et al., 2023a). As a result, any attempt to learn the secret key and watermark would require breaking the underlying cryptographic hash function. On the other hand, if an adversary forges a pair of counterfeit secret key and watermark through reverse engineering without considering the hash mapping relationship, the adversary may achieve a watermark detection rate exceeding the security threshold $\rho^*$ but will fail to satisfy the hash mapping relationship. However, the legitimate model owner can present a valid pair of secret key and watermark that not only exceeds $\rho^*$, but also satisfies the hash mapping relationship. As established in Theorem 1, the probability of such an occurrence occurring by chance is negligible, providing strong cryptographic evidence to support third-party verification agencies in correctly determining the model's ownership. In summary, forging attacks through reverse engineering in NeuralMark is infeasible, regardless of whether the hash mapping relationship is considered.

## B.3 COMPARISON WITH RELATED STUDIES

We compare NeuralMark with several existing studies. To our humble knowledge, the most closely related watermarking methods are presented in (Uchida et al., 2017), (Liu et al., 2021), and (Li et al., 2024), referred to as VanillaMark, GreedyMark, and VoteMark, respectively. VanillaMark serves as the foundation for GreedyMark, VoteMark, and NeuralMark, but NeuralMark substantially differs from them in the following aspects. (1) VanillaMark relies solely on the average pooling mechanism to resist fine-tuning and pruning attacks, but it is ineffective against forging and overwriting attacks. (2) Although GreedyMark selects important parameters for watermark embedding and verification, it fails to effectively resist overwriting attacks with varying attack strengths, such as different values of the hyper-parameter $\lambda$ and the learning rate $\eta$ (see details in Table 7). (3) VoteMark incorporates a random noise mechanism for watermark embedding and verification, which improves robustness to a certain extent, but it remains ineffective against forging and overwriting attacks.

## B.4 LIMITATIONS AND BROADER IMPACT

Although NeuralMark demonstrates promising results and can be seamlessly integrated into various architectures, it has certain limitations. Specifically, it requires direct access to the model parameters, making it unsuitable for verifying ownership through a remote Application Programming Interface (API) where model parameters remain inaccessible. To address this limitation, a potential solution involves integrating NeuralMark with black-box NNW watermarking methods, such as those proposed in (Fan et al., 2019; 2021). Specifically, trigger samples can be utilized alongside vanilla training samples to train the model while embedding the watermark through NeuralMark. This method enables the initial verification of model ownership by evaluating the prediction performance of trigger samples via the remote API. Based on this preliminary evidence, a formal request can be made to the API service provider for access to the corresponding model parameters. Once obtained, NeuralMark can be employed for a secondary, white-box verification to conclusively confirm model ownership. The practical implementation of this combined method is beyond the scope of this work and will be explored in future research.

Ownership protection of artificial intelligence models is a critical and pressing issue. This paper presents a simple yet general method to safeguard model ownership. Our work aims to inspire further academic research in this vital area and advance industry adoption to effectively address ownership concerns related to models.

## C  PROOF FOR THEOREM 1

**Theorem 1** *Under the assumption that the hash function produces uniformly distributed outputs (Bellare & Rogaway, 1993), for a model watermarked by NeuralMark with a watermark tuple $\{\mathbf{K}, \mathbf{b}\}$, where $\mathbf{b} = \mathcal{H}(\mathbf{K})$, if an adversary attempts to forge a counterfeit watermark tuple $\{\mathbf{K}', \mathbf{b}'\}$ such that $\mathbf{b}' = \mathcal{H}(\mathbf{K}')$ and $\mathbf{K}' \neq \mathbf{K}$, then the probability of achieving a watermark detection rate of at least $\rho$ (i.e., $\geq \rho$) is upper-bounded by $\frac{1}{2^n} \sum_{i=0}^{n-\lceil \rho n \rceil} \binom{n}{i}$.*

*Proof.* Since the hash function produces uniformly distributed outputs, each bit of the counterfeit watermark matches the corresponding bit of the extracted watermark from model parameters with a probability of $\frac{1}{2}$. The number of matching bits follows a binomial distribution with parameters $n$ and $p = \frac{1}{2}$. To achieve a detection rate of at least $\rho$, the adversary needs at least $\lceil \rho n \rceil$ bits to match out of $n$ bits. Thus, the probability of having at least $\lceil \rho n \rceil$ matching bits is given by

$$\Pr\left[X \geq \lceil \rho n \rceil\right] = \sum_{i=\lceil \rho n \rceil}^{n} \binom{n}{i} \left(\frac{1}{2}\right)^i \left(\frac{1}{2}\right)^{n-i} = \frac{1}{2^n} \sum_{i=\lceil \rho n \rceil}^{n} \binom{n}{i} = \frac{1}{2^n} \sum_{i=0}^{n-\lceil \rho n \rceil} \binom{n}{i}. \quad (4)$$

Accordingly, the probability of an adversary forging a counterfeit watermark that achieves a watermark detection rate of at least $\rho$ (*i.e.*, $\geq \rho$) is upper-bounded by $\frac{1}{2^n} \sum_{i=0}^{n-\lceil \rho n \rceil} \binom{n}{i}$.

## D  IMPLEMENTATION DETAILS

We implement NeuralMark using the PyTorch framework (Paszke et al., 2019) and conduct all experiments on three NVIDIA V100 series GPUs.

For the image classification architectures, we train for 200 epochs with a multi-step learning rate schedule from scratch, with learning rates set to 0.01, 0.001, and 0.0001 for epochs 1 to 100, 101 to 150, and 151 to 200, respectively. We apply a weight decay of $5 \times 10^{-4}$ and set the momentum to 0.9. The batch sizes for the training and test datasets are set to 64 and 128, respectively. In addition, we set hyper-parameter $\lambda$ to 1 and the number of filter rounds $R$ to 4.

For the GPT-2-S and GPT-2-M architectures, we utilize the Low-Rank Adaptation (LoRA) technique (Hu et al., 2022). Each architecture is trained for 5 epochs with a linear learning rate scheduler, starting at $2 \times 10^{-4}$. We set the warm-up steps to 500, apply a weight decay with a coefficient of 0.01, and enable bias correction in the AdamW optimizer (Loshchilov et al., 2017). The dimension and the scaling factor for LoRA are set to 4 and 32, respectively, with a dropout probability of 0.1 for the LoRA layers. The batch sizes for the training and test sets are 8 and 4, respectively. Moreover, we set hyper-parameter $\lambda$ to 1 and the number of filter rounds $R$ to 10.

## E  ADDITIONAL EXPERIMENTAL RESULTS

### E.1  FINE-TUNING ATTACKS AGAINST WATERMARK EMBEDDING LAYER

Table 9 reports the experimental results of fine-tuning the watermark embedding layer and classifier. As can be seen, the watermark detection rate remains at 100%, but the model performance exhibits a substantial decline. Specifically, for the CIFAR-10 to CIFAR-100 task using ResNet-18, the accuracy achieved by NeuralMark through fine-tuning the watermark embedding layer is 49.77%, which is markedly lower than the 71.67% accuracy obtained when all parameters are fine-tuned. Similar trends are observed across other methods. Those results indicate that solely fine-tuning the watermark embedding layer and classifier makes it challenging to ensure effective model performance.

Table 9: Comparison of resistance to fine-tuning attacks against watermark embedding layer using ResNet-18. Values (%) inside and outside the bracket are watermark detection rate and classification accuracy, respectively.

| Fine-tuning | Clean | | NeuralMark | | VanillaMark | | GreedyMark | | VoteMark | |
|---|---|---|---|---|---|---|---|---|---|---|
| | AlexNet | ResNet-18 | AlexNet | ResNet-18 | AlexNet | ResNet-18 | AlexNet | ResNet-18 | AlexNet | ResNet-18 |
| CIFAR-100 to CIFAR-10 | 85.55 | 89.15 | 85.35(100) | 88.83(100) | 85.48(91.01) | 89.35(85.93) | 80.41(96.48) | 76.15(94.14) | 84.97(89.06) | 89.66(85.54) |
| CIFAR-10 to CIFAR-100 | 58.96 | 49.74 | 58.50(100) | 49.77(100) | 58.75(74.21) | 49.97(70.31) | 51.75(97.65) | 19.94(82.42) | 58.81(80.07) | 49.08(71.87) |
| Caltech-256 to Caltech-101 | 47.65 | 74.09 | 71.29(100) | 73.12(100) | 71.56(100) | 74.03(100) | 72.04(100) | 68.45(100) | 71.62(100) | 72.47(99.60) |
| Caltech-101 to Caltech-256 | 40.61 | 40.00 | 40.34(100) | 40.34(100) | 40.71(96.09) | 39.04(93.36) | 40.68(100) | 36.45(98.82) | 39.52(95.31) | 39.73(93.75) |

## E.2 PRUNING ATTACKS

Figure 6-8 provide additional results from pruning attacks conducted on the CIFAR-10, Caltech-101, and Caltech-256 datasets, respectively. We observe similar trends as those exhibited on the CIFAR-100 dataset, as depicted in Figure 3. Those results collectively suggest NeuralMark exhibits superior robustness in resisting pruning attacks compared to other methods.

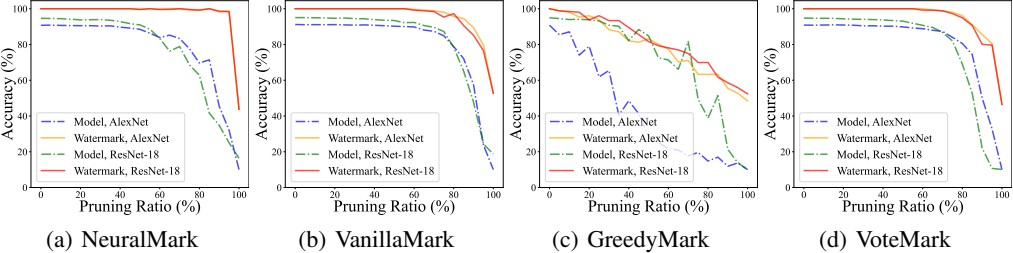

|            (a) NeuralMark            (b) VanillaMark            (c) GreedyMark            (d) VoteMark

Figure 6: Comparison of resistance to pruning attacks at various pruning ratios on the CIFAR-10 dataset using AlexNet and ResNet-18, respectively.

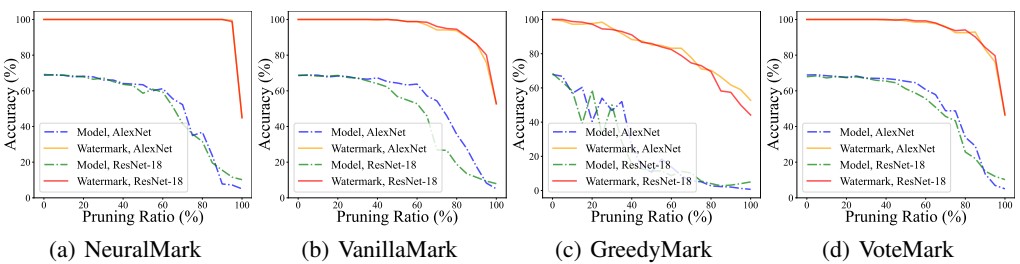

|            (a) NeuralMark            (b) VanillaMark            (c) GreedyMark            (d) VoteMark

Figure 7: Comparison of resistance to pruning attacks at various pruning ratios on the Caltech-101 dataset using AlexNet and ResNet-18, respectively.

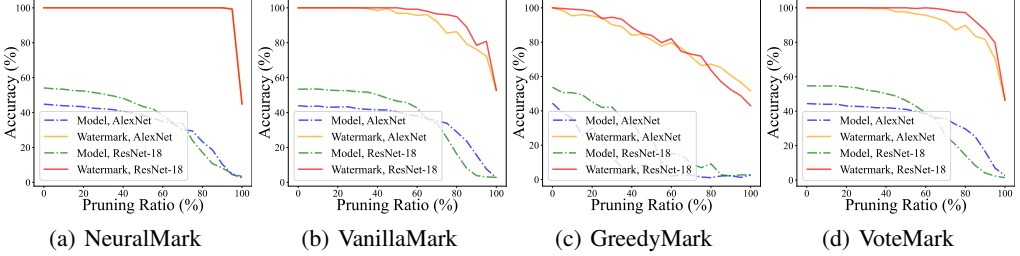

|            (a) NeuralMark            (b) VanillaMark            (c) GreedyMark            (d) VoteMark

Figure 8: Comparison of resistance to pruning attacks at various pruning ratios on the Caltech-256 dataset using AlexNet and ResNet-18, respectively.

## E.3 PRUNING + FORGING ATTACKS WITH DISTINCT PRUNING RATIOS

Table 10 lists more forging attack results with various pruning ratios. As can be seen, NeuralMark can effectively resist forging attacks regardless of the pruning ratio. This is because NeuralMark establishes a hash mapping between the secret key and the watermark, ensuring that its ability to resist forging attacks is not affected by parameter pruning.

Table 10: Comparison of resistance to pruning + forging attacks with distinct pruning ratios on the CIFAR-100 dataset using ResNet-18.

| Pruning Ratio | NeuralMark | VanillaMark | GreedyMark | VoteMark |
|---|---|---|---|---|
| 20% | 49.57 | 100.00 | 50.43 | 100.00 |
| 40% | 49.37 | 100.00 | 50.27 | 100.00 |
| 60% | 52.11 | 100.00 | 47.97 | 100.00 |
| 80% | 50.94 | 100.00 | 49.45 | 100.00 |

### E.4 PARAMETER DISTRIBUTION

Figure 9 provides additional parameter distributions for various architectures on the CIFAR-100 dataset. As can be seen, the parameter distributions of Clean and NeuralMark closely align in each architecture. Those results further demonstrate the secrecy of NeuralMark.

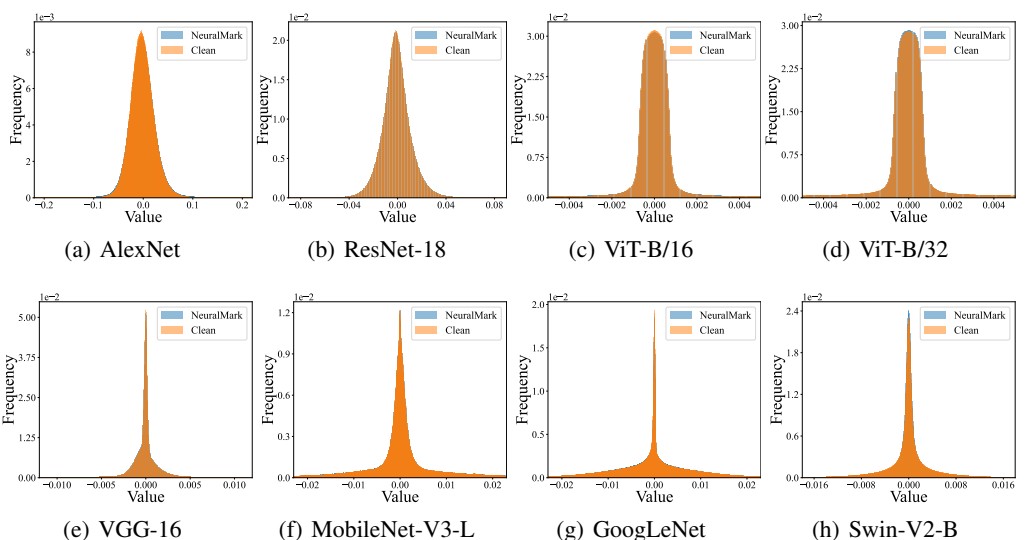

Figure 9: Comparison of parameter distributions with distinct architectures on the CIFAR-100 dataset.

### E.5 PERFORMANCE CONVERGENCE

Figure 10 presents additional performance convergence plots for various architectures on the CIFAR-100 dataset. Across all architectures, the performance curves of Clean and NeuralMark exhibit similar trends and are closely aligned, further confirming that NeuralMark does not negatively affect performance convergence.

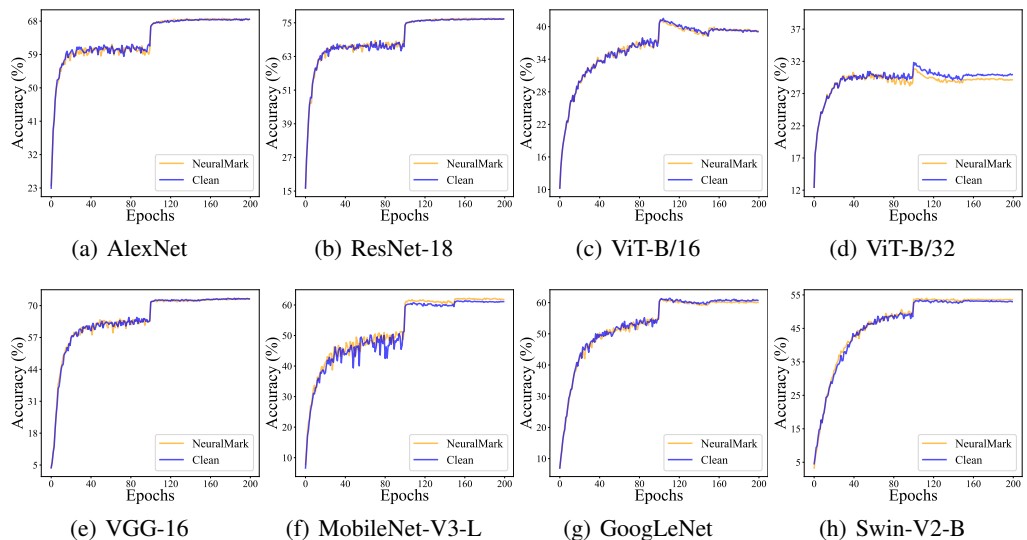

Figure 10: Comparison of performance convergences with distinct architectures on the CIFAR-100 dataset.

## E.6 FILTERING ROUNDS

To assess the influence of the number of filtering rounds on NeuralMark's robustness in resisting various attacks, we conduct additional experiments using 6 and 8 filters, compared to NeuralMark's default setting of 4 filters. We omit forging attacks as the hash mapping mechanism is orthogonal to the watermark filtering process.

Table 11 presents the impact of watermark embedding on the model performance across distinct filtering rounds. The results demonstrate that NeuralMark, even with varying filtering rounds, has a minimal effect on model performance while successfully embedding watermarks.

Table 11: Comparison of classification accuracy (%) with various distinct filter rounds on the CIFAR-10 and CIFAR-100 datasets using ResNet-18, respectively. Watermark detection rates are omitted as they all reach 100%.

| Dataset | 4 Filters | 6 Filters | 8 Filters |
|---------|-----------|-----------|-----------|
| CIFAR-10 | 94.79 | 94.74 | 94.88 |
| CIFAR-100 | 76.74 | 75.59 | 76.16 |

Table 12 reports the results of fine-tuning attacks across distinct filtering rounds. We can observe that NeuralMark maintains a watermark detection rate of 100% across all filtering rounds, with negligible impact on model performance.

Table 12: Comparison of resistance to fine-tuning attacks with distinct filter rounds using ResNet-18. Watermark detection rates are omitted as they all reach 100%.

| Fine-tuning | Clean | 4 Filters | 6 Filters | 8 Filters |
|-------------|-------|-----------|-----------|-----------|
| CIFAR-100 to CIFAR-10 | 93.21 | 93.74 | 93.01 | 93.55 |
| CIFAR-10 to CIFAR-100 | 72.17 | 71.67 | 72.68 | 72.27 |

Figure 11 shows the results of pruning attacks across different filtering rounds. As can be seen, as the number of filtering rounds increases, the robustness of NeuralMark in resisting pruning attacks exhibits a slight decline. One reason is that increasing the number of filter rounds reduces the number of parameters, leading to a smaller average pooling window size, which affects the robustness against pruning attacks to some extent.

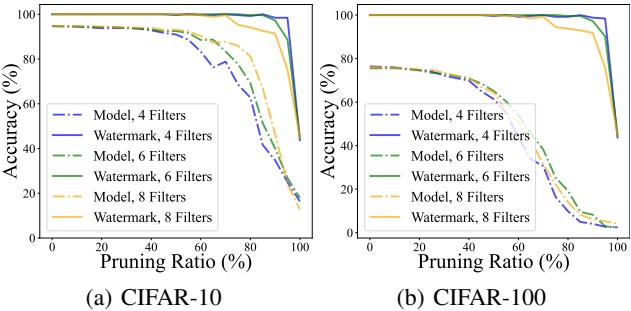

(a) CIFAR-10        (b) CIFAR-100

Figure 11: Comparison of resistance to pruning attacks with distinct filter rounds on the CIFAR-10 and CIFAR-100 datasets using ResNet-18 at various pruning ratios.

Table 13 lists the results of overwriting attacks across distinct filtering rounds. From the results, we find that when the number of filtering rounds is set to 6, NeuralMark exhibits superior robustness compared to 4 and 8 filter rounds. Specifically, at $\eta = 0.01$, the original watermark detection rates for 4, 6, and 8 filter rounds are 92.18%, 94.92%, and 89.84%, respectively. Those results indicate that increasing the number of filtering rounds can enhance robustness against overwriting attacks to a certain extent. However, when the number of filtering rounds exceeds a certain threshold, the robustness may be slightly compromised due to the reduction in the number of parameters.

In summary, NeuralMark maintains its robustness even as the number of filtering rounds increases.

Table 13: Comparison of resistance to overwriting attacks at various trade-off hyper-parameters ($\lambda$) and learning rates ($\eta$) with distinct filtering rounds using ResNet-18. Values (%) inside and outside the bracket are watermark detection rate and classification accuracy, respectively.

| Overwriting | $\lambda$ | 4 Filters | 6 Filters | 8 Filters | $\eta$ | 4 Filters | 6 Filters | 8 Filters |
|---|---|---|---|---|---|---|---|---|
| CIFAR-100 to CIFAR-10 | 1 | 93.65 (100) | 93.13(100) | 93.40(100) | 0.001 | 93.65 (100) | 93.13(100) | 93.40(100) |
| | 10 | 93.44 (100) | 93.06(100) | 93.41(100) | 0.005 | 91.76 (99.60) | 92.10(100) | 91.62(100) |
| | 50 | 93.46 (100) | 93.06(100) | 93.54(100) | 0.01 | 91.58 (92.18) | 91.64(94.92) | 90.48(89.84) |
| | 100 | 93.53 (100) | 92.88(100) | 92.99(100) | 0.1 | 75.2 (50.78) | 75.84(58.2) | 74.54(51.56) |
| | 1000 | 93.09 (100) | 93.03(100) | 93.39(100) | 1 | 10.00 (44.53) | 10.00(47.26) | 10.00(50.39) |
| CIFAR-10 to CIFAR-100 | 1 | 71.78 (100) | 71.69(100) | 72.63(100) | 0.001 | 71.78 (100) | 71.69(100) | 72.63(100) |
| | 10 | 72.6 (100) | 72.06(100) | 72.81(100) | 0.005 | 71.04 (99.60) | 70.65(100) | 71.46(100) |
| | 50 | 72.73 (100) | 71.85(100) | 72.85(100) | 0.01 | 69.14 (96.48) | 69.47(97.26) | 67.88(95.70) |
| | 100 | 71.49 (100) | 71.88(100) | 72.00(100) | 0.1 | 51.88 (60.54) | 55.18(62.10) | 50.36(55.07) |
| | 1000 | 71.81 (100) | 72.22(100) | 72.39(100) | 1 | 1.00 (44.53) | 1.00(47.26) | 1.00(50.39) |

## E.7 WATERMARKING LAYERS

To investigate the impact of watermark embedding layers on model performance, we randomly choose four individual layers and all layers from ResNet-18 for watermark embedding. Table 14 presents the results on the CIFAR-100 dataset, showing that embedding different layers or all layers does not significantly affect model performance.

Table 14: Comparison of classification accuracy (%) on different watermarking layers on the CIFAR-100 dataset using ResNet-18. Here, Layers 1-4 denote randomly chosen layers, while All Layer refers to all layers. Watermark detection rates are omitted as they all reach 100%.

| Watermarking Layer | Layer 1 | Layer 2 | Layer 3 | Layer 4 | All Layer |
|---|---|---|---|---|---|
| Accuracy | 76.51 | 76.68 | 76.30 | 76.73 | 75.86 |

## E.8 WATERMARK LENGTH

To evaluate the influence of watermark length on model performance, we set watermark lengths to 64, 128, 256, 512, 1024, and 2048, respectively. Table 15 illustrates the results on the CIFAR-100 dataset, indicating that NeuralMark achieves a 100% detection rate with various watermark lengths while maintaining nearly lossless model performance.

Table 15: Comparison of classification accuracy (%) for distinct watermark lengths on the CIFAR-100 dataset using ResNet-18. Watermark detection rates are omitted as they all reach 100%.

| Watermark Length | 64 | 128 | 256 | 512 | 1024 | 2048 |
|---|---|---|---|---|---|---|
| Accuracy | 75.84 | 75.90 | 76.46 | 76.18 | 76.51 | 76.27 |

## E.9 TRAINING EFFICIENCY

In Table 16, we report the average time cost (in seconds) per training epoch over five epochs on the CIFAR-100 dataset using ResNet-18. NeuralMark's running time is comparable to that of Clean and VanillaMark, highlighting the efficiency of both watermark filtering and average pooling. Also, NeuralMark significantly outperforms GreedyMark in terms of speed due to GreedyMark's reliance on costly sorting operations for parameter selection, which NeuralMark avoids. NeuralMark demonstrates significantly faster running times compared to VoteMark, as it avoids the multiple rounds of watermark embedding loss calculations required by VoteMark. Those results highlight the superior efficiency of NeuralMark.

Table 16: Comparison of average time cost (in seconds) on the CIFAR-100 dataset using ResNet-18. Here, $R$ denotes the number of filtering rounds.

| Method | Clean | NeuralMark ($R=1$) | NeuralMark ($R=2$) | NeuralMark ($R=3$) | NeuralMark ($R=4$) | VanillaMark | GreedyMark | VoteMark |
|---|---|---|---|---|---|---|---|---|
| Time (s) | 23.60 | 24.49 | 24.94 | 25.01 | 25.19 | 24.34 | 47.43 | 35.17 |

