# OpenReview forum: "NeuralMark: Advancing White-Box Neural Network Watermarking"
_ICLR.cc/2025/Conference — ICLR 2025 Conference Withdrawn Submission_

### Official Review · Reviewer_2mAv · 2024-10-27

**Soundness:** 3
**Presentation:** 3
**Contribution:** 2
**Rating:** 6
**Confidence:** 3

**Summary:**

This paper proposes an advanced white-box watermarking technique that defends against watermark removal attacks through average pooling, establishes a key-to-watermark hash mapping to defend against watermark forgery attacks, and also enhances watermark privacy by using binarised watermarks as a filter for watermark injection parameter selection.

**Strengths:**

1. it makes sense to use watermarking by binarisation as a filter for the injected parameters.
2. the authors provide a theoretical analysis of the security bounds of NeuralMark.
3. extensive experiments show that NeuralMark can effectively resist watermarking attacks without significantly affecting the model.

**Weaknesses:**

The authors establish a key-to-watermark hash mapping, but a similar method for establishing a passport sample-to-watermark hash mapping is already available in [1]. Despite the author's citation of the article in [1], the article does not explain the necessity of using keys directly for hash mapping. In addition, to defend against Removal+Forging Attack, the authors use average pooling, which was also proposed in previous articles [2,3]. As a key module to defend against watermark removal attacks, the approach used by the authors does not advance the knowledge in the field of watermarking.

[1]Hanwen Liu, Zhenyu Weng, Yuesheng Zhu, and Yadong Mu. 2023. Trapdoor normalization with irreversible ownership verification. In Proceedings of the 40th International Conference on Machine Learning (ICML'23).
[2]Yusuke Uchida, Yuki Nagai, Shigeyuki Sakazawa, and Shin'ichi Satoh. 2017. Embedding Watermarks into Deep Neural Networks. In Proceedings of the 2017 ACM on International Conference on Multimedia Retrieval (ICMR '17).
[3] Liu, H., Weng, Z., & Zhu, Y. (2021). Watermarking Deep Neural Networks with Greedy Residuals. International Conference on Machine Learning.

**Questions:**

Three levels of watermarking attacks are presented in Section 2.3.2. However, the Threat Model in Section 3.3 does not seem to be analysed in conjunction with the levels of watermarking attacks.The Removal+Forging Attack first attempts a watermark removal attack, which belongs to Level II in Section 3.2. This is followed by the Forging Attack. I'm not sure if it becomes more difficult to perform a forging attack on a model that has already had its watermark removed. If the attack is successful, this seems to be just a combination of Level I and Level II. It seems like it would look more reasonable to switch the order of the two attacks.

 For the experiments of watermark removal by pruning in Removal+Forging Attack, the author chooses to carry out the watermark forging attack with 40% pruning rate, combining with Figure3 to see that 40% doesn't seem to be a special turning point, and I'm very curious about the robustness of NeuralMark's other different pruning rates to watermark forging attacks. .

---

> ### Author Response · Authors · 2024-11-24
> **Rebuttal-Part I**
>
> Thank you for your thoughtful review and valuable feedback. We address your concerns as follows.
>
> > W1. The authors establish a key-to-watermark hash mapping, but a similar method for establishing a passport sample-to-watermark hash mapping is already available in [1]. Despite the author's citation of the article in [1], the article does not explain the necessity of using keys directly for hash mapping. In addition, to defend against Removal+Forging Attack, the authors use average pooling, which was also proposed in previous articles [2,3]. As a key module to defend against watermark removal attacks, the approach used by the authors does not advance the knowledge in the field of watermarking.
>
> **AW1:** We are sorry for this confusion about the key-to-watermark hash mapping, and we address your concerns as follows.
>
> (1) **We propose a direct hash binding of the secret key with the watermark to defend against forging attacks** (Please refer to **AW1** addressed to **Reviewer ZqfG** for a detailed explanation). Although [1] establishes a hash mapping between passport samples and the watermark, it still necessitates replacing the normalization layer parameters (e.g., batch normalization) with those generated from passport samples. This significantly complicates deployment in practical scenarios. Also, the primary purpose of replacing normalization layer parameters in passport-based methods is to defend against forging attacks. However, this can also be effectively achieved through the hash mapping between passport samples and the watermark. Thus, **the use of both mechanisms within a single method is redundant**. In contrast, **NeuralMark simplifies the watermarking design by directly establishing a hash mapping between the secret key and the watermark, eliminating the need for additional passport samples**. This streamlined approach reduces complexity, making NeuralMark simpler and more practical.
>
> (2) **We propose a watermark filtering mechanism to resist overwriting attacks with distinct strengths**: Most white-box watermarking methods assume that the strength of the adversary's overwriting attack is equivalent to that of the original watermark embedding. **This assumption, however, is not reasonable because adversaries cannot be limited in the strength of their overwriting attack**. To resist overwriting attacks of varying strengths, we propose a watermark filtering mechanism. This mechanism utilizes the watermark as a filter to select model parameters for embedding, making it significantly increases the difficulty for adversaries to ascertain and manipulate those parameters. As a result, **this mechanism effectively reduces interference with the original watermark (Strength 2 from 33YR and Strength 1 from Reviewer 2mAr), even when adversaries increase the embedding strength of their own watermark**. To the best of our knowledge, there is no existing method that utilizes the watermark as a filter for selecting model parameters to resist overwriting attacks of varying strengths.
>
> (3) **We incorporate hash mapping, watermarking filtering, and average pooling mechanisms into a unified method**: Although average pooling has been used in previous works [2, 3] to defend against removal attacks, **our use of this technique is integrated into a unified method that includes hash mapping and watermark filtering mechanisms**. This integration ensures a robust defense against watermark removal, forging, and overwriting attacks while maintaining practical deployment. To our humble knowledge, NeuralMark is the first to incorporate hash mapping, watermarking filtering, and average pooling mechanisms into a unified method. Also, we provide a theoretical analysis of NeuralMark's security boundary.
>
> (4) **We extend white-box model watermarks to the Transformer architectures (** *i.e.*, **ViT, GPT-2 ), whereas previous work does not** (**Strength 2** from **Reviewer 464P**): NeuralMark can be seamlessly integrated into various neural network architectures due to its simplicity and practicality, which are crucial for advancing the development and widespread adoption of white-box watermarking methods.
>
> In summary, we believe that our work makes meaningful contributions to the white-box watermarking field, addressing the key limitations and advancing the state of the art. We hope this explanation clarifies the value and impact of our contributions.
>
> [1] Hanwen Liu, Zhenyu Weng, Yuesheng Zhu, and Yadong Mu. 2023. Trapdoor normalization with irreversible ownership verification. ICML’23.
>
> [2] Yusuke Uchida, Yuki Nagai, Shigeyuki Sakazawa, and Shin’ichi Satoh. 2017. Embedding Watermarks into Deep Neural Networks. ICMR ’17.
>
> [3] Liu, H., Weng, Z., Zhu, Y. (2021). Watermarking Deep Neural Networks with Greedy Residuals. ICML’21.

---

> > ### Author Response · Authors · 2024-11-24
> > **Rebuttal-Part II**
> >
> > > Q1. Three levels of watermarking attacks are presented in Section 2.3.2. However, the Threat Model in Section 3.3 does not seem to be analysed in conjunction with the levels of watermarking attacks. The Removal+Forging Attack first attempts a watermark removal attack, which belongs to Level II in Section 3.2. This is followed by the Forging Attack. I'm not sure if it becomes more difficult to perform a forging attack on a model that has already had its watermark removed. If the attack is successful, this seems to be just a combination of Level I and Level II. It seems like it would look more reasonable to switch the order of the two attacks.
> >
> > **AQ1:** Thanks for your insightful suggestion. **As explained in AW1 addressed to Reviewer ZqfG, NeuralMark's resilience to forging attacks is fundamentally underpinned by the key-to-watermark hash mapping**. Consequently, **regardless of the order of operations, NeuralMark can effectively resist forging attacks**. For this reason, we do not separate the analysis of the forging attack in the manuscript. To address your concern and ensure clarity, we have reorganized the presentation of the attack types to better align with the levels of success criteria for watermarking attacks:
> > >Based on the defined success criteria for watermarking attacks, the adversary can launch the following attacks. (1) Forging Attack: the adversary performs forging attacks to forge a pair of counterfeit secret key and watermark without altering the model parameters. Specifically, we employ reverse engineering attacks [1-2], which involve randomly forging a counterfeit watermark and subsequently deriving a corresponding secret key by freezing the model parameters.
> >
> > Furthermore, we have provided empirical verification of forging attacks alone. The results are listed in Table 1. As can be seen, the counterfeit watermark detection rate of NeuralMark remains near 50%, akin to a random guess. This demonstrates the robustness of NeuralMark against forging attacks. Please refer to **Forging Attack** in `Section 5.3` in the revision for the detailed changes and experimental results.
> >
> > **Table 1. Comparison of resistance to forging attacks using ResNet-18**
> > | Dataset    | NeuralMark | VanillaMark | GreedyMark | VoteMark |
> > |------------|------------|-------------|------------|----------|
> > | CIFAR-10   | 48.56      | 100.00      | 50.70      | 100.00   |
> > | CIFAR-100  | 49.41      | 100.00      | 52.85      | 100.00   |
> >
> > [1] Lixin Fan, Kam Woh Ng, and Chee Seng Chan. Rethinking deep neural network ownership verification: Embedding passports to defeat ambiguity attacks. In NeurIPS, volume 32, 2019.
> >
> > [2] Lixin Fan, Kam Woh Ng, Chee Seng Chan, and Qiang Yang. Deepipr: Deep neural network ownership verification with passports. IEEE Transactions on Pattern Analysis and Machine Intelligence, 44 (10):6122–6139, 2021.
> >
> > > Q2. For the experiments of watermark removal by pruning in Removal+Forging Attack, the author chooses to carry out the watermark forging attack with 40% pruning rate, combining with Figure3 to see that 40% doesn't seem to be a special turning point, and I'm very curious about the robustness of NeuralMark's other different pruning rates to watermark forging attacks.
> >
> > **AQ2:** We are thankful for your thoughtful feedback. **As explained in AW1 addressed to Reviewer ZqfG, NeuralMark's resilience to forging attacks is fundamentally underpinned by the key-to-watermark hash mapping**. Consequently, **regardless of the pruning ratio applied, NeuralMark can effectively resist forging attacks**.
> >
> > To address your concern about pruning ratios, we have conducted experiments to evaluate NeuralMark's robustness under varying pruning rates (*i.e*., 20%, 40%, 60%, and 80%). Since the figures cannot be displayed in rebuttal, we list some key results in the following table. Table 2 show that the watermark detection rate of NeuralMark remained close to 50% with various pruning ratios, similar to random guessing. We have included the results in the revision, please refer to `Appendix E.3` for details.
> >
> > **Table 2. Comparison of resistance to pruning + forging attacks with distinct pruning ratios on the CIFAR-100 dataset using ResNet-18**
> > | Pruning Ratio    | NeuralMark | VanillaMark | GreedyMark | VoteMark |
> > |------------|------------|-------------|------------|----------|
> > | 20%   | 49.57      | 100.00      | 50.43      | 100.00   |
> > | 40%  | 49.37      | 100.00      | 50.27      | 100.00   |
> > | 60%  | 52.11      | 100.00      | 47.97      | 100.00   |
> > | 80%  | 50.94      | 100.00      | 49.45      | 100.00   |

---

> > > ### Author Response · Authors · 2024-11-30
> > > **Request for Reconsideration of the score**
> > >
> > > Dear Reviewer 2mAv
> > >
> > > We sincerely appreciate the insightful and constructive feedback you have provided. We have made a thorough and diligent effort to address all of your concerns, and we believe these revisions have significantly enhanced the quality of our manuscript. **We are deeply grateful for your valuable suggestions, and if, after reviewing the revised manuscript, you feel it now meets your expectations, we would be sincerely thankful if you could consider raising your score**.
> > >
> > > Should you have any further questions or require additional clarifications, please do not hesitate to reach out, and we would be happy to assist before the rebuttal deadline.
> > >
> > > Best regards
> > >
> > > Authors

---

> > > > ### Author Response · Authors · 2024-12-03
> > > > **Request for Final Feedback**
> > > >
> > > > Dear Reviewer 2mAv:
> > > >
> > > > Thank you once again for your thoughtful and constructive review of our manuscript. We sincerely hope that all reviewers will approach the entire review process with the utmost responsibility. **We would greatly appreciate it if you could provide your final scores and any additional comments before the rebuttal period concludes**. Many thanks.
> > > >
> > > > Best regards
> > > >
> > > > Authors

---

### Official Review · Reviewer_464P · 2024-10-30

**Soundness:** 2
**Presentation:** 3
**Contribution:** 2
**Rating:** 5
**Confidence:** 4

**Summary:**

The paper introduces NeuralMark, a method for white-box neural network watermarking to protect ownership. It integrates seamlessly into various network architectures, using hash mapping, watermark filtering, and average pooling to enhance security against forging, overwriting, fine-tuning, and pruning attacks. NeuralMark is empirically validated across 14 different architectures and multiple tasks, demonstrating its effectiveness while maintaining model performance and security.

**Strengths:**

- The research direction about model watermarks is important. As deep learning is driven by the scale, more and more resources have been invested in designing and developing deep models, and therefore ownership protection methods are necessary for safeguarding these models of interests.
- This paper extended model watermarks to the Transformer architecture, while previous work didn't. Empirical results on Vision Transformers,  GPT-2 and Swin Transformers demonstrate the effectiveness of the proposed method.
- The presentation is clear and in general well-written.

**Weaknesses:**

- The proposed method seems to be incremental. I am aware that there are some numerical improvements in the reported results. However, no substantial contribution to watermark design, in the aspects of robustness or fidelity for example.
- Further results on ambiguity attacks (i.e., attacks that would cast ambiguity over the verification process, including forging attacks and overwriting attacks) are needed. See below for details.

**Questions:**

- To my understanding, the proposed algorithm is public while the watermark (i.e., the specific parameter filter in the proposed method) is secret, according to Kerckhoff's Principle. After preliminary verification, the watermark is also publically available, and since the adversary also knows the hidden watermark, what happens to the next verification process? After the first verification, anyone knows the embedded watermark and anyone also could claim ownership, which may cast ambiguity over the verification process after the first verification. In other words, can the proposed method support public verification?
- For the above question, I think one possible solution is to map the owner's signature to the watermark binary code, e.g.,  [1, 0, 1, 0]. If so, I would suggest additional analyses and ablation studies about the binary code design in the experiment section.
- Will the source codes and datasets with necessary documents and instructions be publicly available?

I would raise my rating if these concerns are addressed.

---

> ### Author Response · Authors · 2024-11-24
> **Rebuttal-Part I**
>
> Thank you for your thoughtful review and valuable feedback. We address your concerns as follows.
>
> > W1. The proposed method seems to be incremental. I am aware that there are some numerical improvements in the reported results. However, no substantial contribution to watermark design, in the aspects of robustness or fidelity for example.
>
> **AW1:** We feel that there may be a misunderstanding regarding the contributions of NeuralMark, which provides the following key contributions to the white-box watermarking field:
>
> (1) **We propose a direct hash binding of the secret key with the watermark to defend against forging attacks** (Please refer to **AW1** addressed to **Reviewer ZqfG** for a detailed explanation): Although [1] establishes a hash mapping between passport samples and the watermark, it still necessitates replacing the normalization layer parameters (e.g., batch normalization) with those generated from passport samples. This significantly complicates deployment in practical scenarios. Also, the primary purpose of replacing normalization layer parameters in passport-based methods is to defend against forging attacks. However, this can also be effectively achieved through the hash mapping between passport samples and the watermark. Thus, **the use of both mechanisms within a single method is redundant**. In contrast, **NeuralMark simplifies the watermarking design by directly establishing a hash mapping between the secret key and the watermark, eliminating the need for additional passport samples**. This streamlined approach reduces complexity, making NeuralMark simpler and more practical.
>
> (2) **We propose a watermark filtering mechanism to resist overwriting attacks with distinct strength levels**: Most white-box watermarking methods assume that the level of strength of the adversary's overwriting attack is equivalent to that of the original watermark embedding. **This assumption, however, is not reasonable because adversaries cannot be limited in the strength of their overwriting attacks**. To resist overwriting attacks of varying strength levels, we propose a watermark filtering mechanism. This mechanism utilizes the watermark as a filter to select model parameters for embedding, making it significantly increase the difficulty for adversaries to ascertain and manipulate those parameters. As a result, **this mechanism effectively reduces interference with the original watermark (Strength 2 from 33YR and Strength 1 from Reviewer 2mAr), even when adversaries increase the embedding strength of their own watermark**. To the best of our knowledge, there is no existing method that utilizes the watermark as a filter for selecting model parameters to resist overwriting attacks of varying strength levels.
>
> (3) **We incorporate hash mapping, watermarking filtering, and average pooling mechanisms into a unified method**: Although average pooling has been used in previous works [2, 3] to defend against
> removal attacks, **our use of this technique is integrated into a unified method that includes hash mapping and watermark filtering mechanisms**. This integration ensures a robust defense against watermark removal, forging, and overwriting attacks while maintaining practical deployment. To our humble knowledge, NeuralMark is the first to incorporate hash mapping, watermarking filtering, and average pooling mechanisms into a unified method. Also, we provide a theoretical analysis of NeuralMark's security boundary.
>
> (4) **We extend white-box model watermarks to the Transformer architectures (** *i.e.,* **ViT, GPT-2 ), whereas previous work does not** (**Strength 2** from **Reviewer 464P**): NeuralMark can be seamlessly integrated into various neural network architectures due to its simplicity and practicality, which are crucial for advancing the development and widespread adoption of white-box watermarking methods.
>
> In summary, we believe that our work makes meaningful contributions to the white-box watermarking field, addressing the key limitations and advancing the state of the art. We hope this explanation clarifies the value and impact of our contributions.
>
> [1] Hanwen Liu, Zhenyu Weng, Yuesheng Zhu, and Yadong Mu. 2023. Trapdoor normalization with irreversible ownership verification. ICML’23.
>
> [2] Yusuke Uchida, Yuki Nagai, Shigeyuki Sakazawa, and Shin’ichi Satoh. 2017. Embedding Watermarks into Deep Neural Networks. ICMR ’17.
>
> [3] Liu, H., Weng, Z., Zhu, Y. (2021). Watermarking Deep Neural Networks with Greedy Residuals. ICML’21.

---

> ### Author Response · Authors · 2024-11-24
> **Rebuttal-Part II**
>
> > W2. Further results on ambiguity attacks (i.e., attacks that would cast ambiguity over the verification process, including forging attacks and overwriting attacks) are needed. See below for details. To my understanding, the proposed algorithm is public while the watermark (i.e., the specific parameter filter in the proposed method) is secret, according to Kerckhoff's Principle. After preliminary verification, the watermark is also publically available, and since the adversary also knows the hidden watermark, what happens to the next verification process? After the first verification, anyone knows the embedded watermark and anyone also could claim ownership, which may cast ambiguity over the verification process after the first verification. In other words, can the proposed method support public verification?
> For the above question, I think one possible solution is to map the owner's signature to the watermark binary code, e.g., [1, 0, 1, 0]. If so, I would suggest additional analyses and ablation studies about the binary code design in the experiment section.
>
> **AW2:** You raise an insightful issue regarding repeated public verifications. In the design of NeuralMark, the identity information of the model owner is not intentionally embedded within the secret key. In practice, however, the model owner is typically the first to perform and publicly verify ownership on the timeline. **This chronological precedence, we believe, provides sufficient evidence to establish ownership**. This is akin to the discovery of a mathematical theorem, *i.e.*, **the first scientist to publicly disclose the theorem is acknowledged as the discoverer, whereas subsequent claims, lacking proof of independent discovery, are not recognized as such**. In contrast, if an adversary attempts to publicly verify ownership of a model before the legitimate model owner, this constitutes forging attacks. However, such attacks are computationally infeasible due to the hash mapping between the secret key and the watermark.
>
> In addition, we appreciate your suggestion, which **has inspired us to design a more practical method capable of supporting repeated public verification by multiple owners**.
> In several practical scenarios where a model is collaboratively developed by multiple owners, their signatures can be seamlessly integrated into NeuralMark to facilitate ownership verification. Specifically, **the signatures of model owners are concatenated with the secret key and then hashed to generate the watermark**, *i.e.*, $\mathbf{b} = \mathcal{H} (\mathbf{S}_1 || \cdots || \mathbf{S}_n || \mathbf{K}) \in \{0, 1\}^n$, where $||$ denotes concatenation operation, and $\mathbf{S}_n$ represents the $n$-th model owner's signature, serving as cryptographic proof of its identity. Accordingly, this mechanism enables repeated public verification by multiple owners.
> Furthermore, its robustness in resisting forging attacks is guaranteed by the cryptographic properties of the hash function, similar to the case where $\mathbf{b} = \mathcal{H} (\mathbf{K})$. Also, **this mechanism is orthogonal to NeuralMark's existing mechanisms (** *i.e.*,**watermark filtering and average pooling) and does not compromise its robustness against other types of attacks**. We have included those discussions in the revision, please refer to `lines 211-212` and `Appendix B.1` for details.
>
> > Q1. Will the source codes and datasets with necessary documents and instructions be publicly available?
>
> **AQ1:** Yes, we will release our source codes and datasets upon acceptance. Indeed, we have already uploaded the source codes in the supplementary materials, and the datasets can be automatically downloaded. Additionally, all the source codes and necessary documents are organized and available at the following anonymous link: https://anonymous.4open.science/r/NeuralMark.

---

> > ### Author Response · Authors · 2024-11-30
> > **Request for Reconsideration of the score**
> >
> > Dear Reviewer 464P
> >
> > We sincerely appreciate the insightful and constructive feedback you have provided. We have made a thorough and diligent effort to address all of your concerns, and we believe these revisions have significantly enhanced the quality of our manuscript. **We are deeply grateful for your valuable suggestions, and if, after reviewing the revised manuscript, you feel it now meets your expectations, we would be sincerely thankful if you could consider raising your score**.
> >
> > Should you have any further questions or require additional clarifications, please do not hesitate to reach out, and we would be happy to assist before the rebuttal deadline.
> >
> > Best regards
> >
> > Authors

---

> > > ### Author Response · Authors · 2024-12-03
> > > **Request for Final Feedback**
> > >
> > > Dear Reviewer 464P:
> > >
> > > Thank you once again for your thoughtful and constructive review of our manuscript. We sincerely hope that all reviewers will approach the entire review process with the utmost responsibility. **We would greatly appreciate it if you could provide your final scores and any additional comments before the rebuttal period concludes**. Many thanks.
> > >
> > > Best regards
> > >
> > > Authors

---

### Official Review · Reviewer_33YR · 2024-10-31

**Soundness:** 3
**Presentation:** 3
**Contribution:** 2
**Rating:** 5
**Confidence:** 4

**Summary:**

The paper introduces NeuralMark, a white-box watermarking method aimed at safeguarding the ownership of deep neural networks by embedding robust watermarks that resist various attacks. NeuralMark employs a three strategies: a hash mapping links a secret key to the watermark, serving as a defense against forging; watermark filtering secures the model parameters from overwriting; and average pooling protects against fine-tuning and pruning attacks.

**Strengths:**

1. NeuralMark integrates hash mapping, which ties the secret key directly to the watermark, thus significantly reducing vulnerability to forgery. This approach makes reverse engineering infeasible by adversaries, as altering the watermark would require breaking the hash.

2. Watermark filtering limits parameter overlap, making overwriting attacks more challenging. This filtering method ensures that even if adversaries attempt to embed a new watermark, the likelihood of interference with the original watermark is minimized.

3. Tests conducted on diverse datasets (e.g., CIFAR, TinyImageNet, and Caltech) show that NeuralMark imposes minimal accuracy loss, indicating that it balances watermark robustness and model fidelity effectively​.

**Weaknesses:**

1. The motivation is unclear. Although the authors explain the motivation for the design of the scheme, they do not explain why they are interested in weight-based watermarking. In the introduction, the author introduces three types of model watermarking techniques and states that all three types of watermarking techniques face the same attacks. It is strange and incomprehensible that the author flatly promotes that they only focus on model weight-based watermarking techniques without clearly comparing the pros and cons of the three types of techniques.

2. The references do not take into account the latest research progress. The introduction of weight-based methods in Related Works only covers work up to 2021. I don't believe there has been no new progress in this direction in the past three years. In addition, in the experimental comparison, only two old solutions were compared, one proposed in 2017 and the other proposed in 2021. There is no comparison with the SOTA solutions, which makes the experimental results unconvincing.

3. This paper says that performing the filtering round more times can reduce the overlap ratio, but it also reduces the number of weight parameters for embedding the watermark. Will this affect the security of the watermark? It is necessary to conduct an experimental analysis.

4. The details of fine-tuning attacks are known.

**Questions:**

None

---

> ### Author Response · Authors · 2024-11-24
> **Rebuttal-Part I**
>
> Thank you for your thoughtful review and valuable feedback. We address your concerns as follows.
>
> > W1. The motivation is unclear. Although the authors explain the motivation for the design of the scheme, they do not explain why they are interested in weight-based watermarking. In the introduction, the author introduces three types of model watermarking techniques and states that all three types of watermarking techniques face the same attacks. It is strange and incomprehensible that the author flatly promotes that they only focus on model weight-based watermarking techniques without clearly comparing the pros and cons of the three types of techniques.
>
> **AW1:** We appreciate your insightful suggestion and have made the following revisions to make the motivation clearer:
>
> * In the `Introduction`, we have provided the motivation behind our interest in weighted-based methods:
>
> > Building on the distinct characteristics of those methods, we are particularly drawn to weight-based methods due to their simplicity and practicability. On one hand, unlike passport-based methods, weight-based methods do not require complex passport layers or incur additional training burdens. On the other hand, unlike activation-based methods, they do not directly constrain the activation maps for watermark embedding. However, the aforementioned limitations of existing weight-based methods motivate us to study the following question: "*How can we design a more effective and robust weight-based NNW method to address those limitations?*"
>
> * In the `Related Work`, we have included the following discussions about the pros and cons of the three types of methods:
>
> > In summary, weight-based methods, while straightforward, often lack robustness against forging and overwriting attacks. Passport-based methods enhance robustness by binding the watermark to model performance but incur significant training overhead and remain vulnerable to overwriting attacks. Similarly, activation-based methods improve robustness by associating the watermark with activation maps, yet they lack flexibility and fail to effectively defend against forging attacks.
>
> Please see details in `lines 048-053` and `108-112` in the revision.

---

> ### Author Response · Authors · 2024-11-24
> **Rebuttal-Part II**
>
> > W2. There is no comparison with the SOTA weight-based solutions, which makes the experimental results unconvincing.
>
> **AW2:** Thank you for your valuable feedback. In our manuscript, we have included two baseline methods: VanillaMark [1], which represents the pioneering work in weight-based methods, and GreedyMark [2], which showcases a representative approach that integrates weight-based methods with cryptographic techniques. Through an extensive literature review, we found that recent solutions combining cryptography with weight-based methods are relatively scarce, which is why we only included VanillaMark and GreedyMark in our comparison experiments. However, other types of weight-based methods are still available. **One recent and representative method [3], published in AAAI'24, presents a unified approach for both white-box and black-box watermarking**. We refer to this method as VoteMark, as it implements a voting-based mechanism for watermark verification, and **we utilize its white-box version for comparison**. Specifically, VoteMark introduces random noises to embed the watermark and then employs a majority voting scheme to aggregate the results of multiple rounds of verification, which can be regarded as an enhanced version of VanillaMark.
>
> Several key results of VoteMark in resisting forging and overwriting attacks are presented in the following tables. As can be seen, VoteMark is less effective than NeuralMark in resisting forging attacks and overwriting attacks with varying strength levels due to its reliance on a majority voting mechanism. Additionally, **VoteMark has been added in the `Introduction` and `Related Work`, and all results have been included in the revision**.
>
> **Table 1. Comparison of resistance to forging attacks using ResNet-18.**
> | Dataset   | NeuralMark | VanillaMark | GreedyMark | VoteMark |
> |-----------|------------|-------------|------------|----------|
> | CIFAR-10  | 48.56      | 100.00      | 50.70      | 100.00   |
> | CIFAR-100 | 49.41      | 100.00      | 52.85      | 100.00   |
>
> **Table 2. Comparison of resistance to overwriting attacks at various trade-off hyper-parameters ($\lambda$) and learning rates ($\eta$). Values (\%) inside and outside the bracket are watermark detection rate and classification accuracy, respectively.**
> | Overwriting       | λ   | NeuralMark         | VanillaMark       | GreedyMark        | VoteMark          | η    | NeuralMark         | VanillaMark       | GreedyMark        | VoteMark          |
> |--------------------|-----|--------------------|-------------------|-------------------|-------------------|-------|--------------------|-------------------|-------------------|-------------------|
> | CIFAR-100 to CIFAR-10 | 1   | 93.65 (100)       | 93.30 (100)       | 93.45 (48.82)     | 93.63 (100)       | 0.001 | 93.65 (100)       | 93.30 (100)       | 93.45 (48.82)     | 93.63 (100)       |
> |                    | 10  | 93.44 (100)       | 93.58 (100)       | 93.29 (51.17)     | 93.13 (100)       | 0.005 | 91.76 (99.60)     | 92.17 (73.04)     | 92.13 (50.00)     | 92.45 (78.90)     |
> |                    | 50  | 93.46 (100)       | 93.50 (100)       | 93.07 (55.07)     | 93.39 (100)       | 0.01  | 91.58 (92.18)     | 91.79 (62.10)     | 91.53 (49.60)     | 91.76 (60.15)     |
> |                    | 100 | 93.53 (100)       | 92.95 (94.53)     | 93.18 (54.29)     | 93.53 (96.48)     | 0.1   | 75.2 (50.78)      | 79.68 (47.26)     | 72.42 (53.12)     | 70.92 (54.29)     |
> |                    | 1000| 93.09 (100)       | 92.89 (53.90)     | 92.85 (49.60)     | 92.77 (59.37)     | 1     | 10.00 (44.53)     | 10.00 (53.51)     | 10.00 (48.04)     | 10.00 (53.51)     |
> | CIFAR-10 to CIFAR-100 | 1   | 71.78 (100)       | 72.68 (98.82)     | 71.34 (55.07)     | 72.97 (98.43)     | 0.001 | 71.78 (100)       | 72.68 (98.82)     | 71.34 (55.07)     | 72.97 (98.43)     |
> |                    | 10  | 72.6 (100)        | 72.03 (98.04)     | 72.30 (49.21)     | 72.08 (98.04)     | 0.005 | 71.04 (99.60)     | 70.02 (69.53)     | 70.25 (48.04)     | 71.11 (71.09)     |
> |                    | 50  | 72.73 (100)       | 72.45 (95.70)     | 70.92 (46.87)     | 72.38 (97.26)     | 0.01  | 69.14 (96.48)     | 69.02 (59.76)     | 69.25 (46.09)     | 68.88 (62.11)     |
> |                    | 100 | 71.49 (100)       | 71.92 (92.18)     | 72.05 (48.04)     | 72.72 (93.75)     | 0.1   | 51.88 (60.54)     | 51.76 (53.90)     | 51.71 (51.56)     | 51.74 (56.25)     |
> |                    | 1000| 71.81 (100)       | 71.35 (57.42)     | 71.74 (51.95)     | 70.73 (56.64)     | 1     | 1.00 (44.53)      | 1.00 (53.15)      | 1.00 (50.00)      | 1.00 (53.51)      |
>
> [1] Yusuke Uchida et al. Embedding Watermarks into Deep Neural Networks. ICMR ’17.
>
> [2] Liu, H. et al. Watermarking Deep Neural Networks with Greedy Residuals. ICML’21.
>
> [3] Fangqi Li et al. Revisiting the information capacity of neural network watermarks: Upper bound estimation and beyond. In AAAI'24.

---

> ### Author Response · Authors · 2024-11-24
> **Rebuttal-Part III**
>
> > W3. This paper says that performing the filtering round more times can reduce the overlap ratio, but it also reduces the number of weight parameters for embedding the watermark. Will this affect the security of the watermark? It is necessary to conduct an experimental analysis.
>
> **AW3:** You raise an interesting point. Performing multiple rounds of filtering reduces the number of watermark embedding parameters. On the one hand, **this reduction decreases the window size of average pooling**, thereby diminishing its effect and, consequently, reducing robustness. On the other hand, since the total number of parameters in the watermark layer remains unchanged, **this reduction makes it more difficult for adversaries to attack the watermark embedding parameters**, thus enhancing robustness. Therefore, **the number of watermark embedding parameters is not strictly positively correlated with robustness**. To address your concern and verify our hypothesis, we have conducted additional experiments using 6 and 8 filters, compared to NeuralMark's default setting of 4 filters. Several key experimental results are listed as follows.
>
> Since the figures cannot be displayed in rebuttal, we list some key results against pruning and overwriting attacks in the following tables. As can be seen, as the number of filtering rounds increases, the robustness of NeuralMark in resisting those attacks exhibits nearly no difference. In summary, NeuralMark maintains its robustness even as the number of filtering rounds increases. Please see details in `Appendix E.6` in the revision.
>
> **Table 3. Comparison of resistance to pruning attacks at various pruning ratios on the CIFAR-100 dataset with distinct filter rounds using ResNet-18.**
> | Pruning Rate | 4 Filters    | 6 Filters    | 8 Filters    |
> |--------------|--------------|--------------|--------------|
> | 10%          | 75.98(100)   | 75.59(100)   | 75.86(100)   |
> | 30%          | 72.18(100)   | 73.14(100)   | 73.44(100)   |
> | 50%          | 61.41(99.61) | 65.12(100)   | 64.46(100)   |
> | 70%          | 30.61(100)   | 38.23(100)   | 31.44(99.22) |
> | 90%          | 4.09(98.82)  | 8.24(97.26)  | 6.15(91.79)  |
>
> **Table 4. Comparison of resistance to overwriting attacks at various trade-off hyper-parameters ($\lambda$) and learning rates ($\eta$) with distinct filtering rounds. Values (\%) inside and outside the bracket are watermark detection rate and classification accuracy, respectively.**
> | Overwriting       | λ   | 4 Filters         | 6 Filters         | 8 Filters         | η    | 4 Filters         | 6 Filters         | 8 Filters         |
> |--------------------|-----|-------------------|-------------------|-------------------|-------|-------------------|-------------------|-------------------|
> | CIFAR-100 to CIFAR-10 | 1   | 93.65 (100)       | 93.13 (100)       | 93.40 (100)       | 0.001 | 93.65 (100)       | 93.13 (100)       | 93.40 (100)       |
> |                    | 10  | 93.44 (100)       | 93.06 (100)       | 93.41 (100)       | 0.005 | 91.76 (99.60)     | 92.10 (100)       | 91.62 (100)       |
> |                    | 50  | 93.46 (100)       | 93.06 (100)       | 93.54 (100)       | 0.01  | 91.58 (92.18)     | 91.64 (94.92)     | 90.48 (89.84)     |
> |                    | 100 | 93.53 (100)       | 92.88 (100)       | 92.99 (100)       | 0.1   | 75.20 (50.78)     | 75.84 (58.20)     | 74.54 (51.56)     |
> |                    | 1000| 93.09 (100)       | 93.03 (100)       | 93.39 (100)       | 1     | 10.00 (44.53)     | 10.00 (47.26)     | 10.00 (50.39)     |
> | CIFAR-10 to CIFAR-100 | 1   | 71.78 (100)       | 71.69 (100)       | 72.63 (100)       | 0.001 | 71.78 (100)       | 71.69 (100)       | 72.63 (100)       |
> |                    | 10  | 72.60 (100)       | 72.06 (100)       | 72.81 (100)       | 0.005 | 71.04 (99.60)     | 70.65 (100)       | 71.46 (100)       |
> |                    | 50  | 72.73 (100)       | 71.85 (100)       | 72.85 (100)       | 0.01  | 69.14 (96.48)     | 69.47 (97.26)     | 67.88 (95.70)     |
> |                    | 100 | 71.49 (100)       | 71.88 (100)       | 72.00 (100)       | 0.1   | 51.88 (60.54)     | 55.18 (62.10)     | 50.36 (55.07)     |
> |                    | 1000| 71.81 (100)       | 72.22 (100)       | 72.39 (100)       | 1     | 1.00 (44.53)      | 1.00 (47.26)      | 1.00 (50.39)      |

---

> > ### Author Response · Authors · 2024-11-24
> > **Rebuttal-Part IV**
> >
> > > W4. The details of fine-tuning attacks are unknown.
> >
> > **AW4:** We are sorry for this confusion. The details of fine-tuning attacks are as follows: Following [1], for all fine-tuning attacks, we use the same hyper-parameters as during training, except for setting the learning rate to 0.001. In our manuscript, we replace the task-specific classifier and minimize the main task loss $\mathcal{L}\_m$ to optimize all parameters for 100 epochs. We have included those details in `lines 372-375` of the revision. In addition, as Revierer ZqfG suggested, we have conducted experiments where only the watermark embedding layer is fine-tuned, please refer to AW2 addressed to Revierer ZqfG for details.
> >
> > [1] Liu, H., Weng, Z., Zhu, Y. (2021). Watermarking Deep Neural Networks with Greedy Residuals. ICML’21.

---

> > > ### Author Response · Authors · 2024-11-30
> > > **Request for Reconsideration of the score**
> > >
> > > Dear Reviewer 33YR
> > >
> > > We sincerely appreciate the insightful and constructive feedback you have provided. We have made a thorough and diligent effort to address all of your concerns, and we believe these revisions have significantly enhanced the quality of our manuscript. **We are deeply grateful for your valuable suggestions, and if, after reviewing the revised manuscript, you feel it now meets your expectations, we would be sincerely thankful if you could consider raising your score**.
> > >
> > > Should you have any further questions or require additional clarifications, please do not hesitate to reach out, and we would be happy to assist before the rebuttal deadline.
> > >
> > > Best regards
> > >
> > > Authors

---

> > > > ### Author Response · Authors · 2024-12-03
> > > > **Request for Final Feedback**
> > > >
> > > > Dear Reviewer 33YR:
> > > >
> > > > Thank you once again for your thoughtful and constructive review of our manuscript. We sincerely hope that all reviewers will approach the entire review process with the utmost responsibility. **We would greatly appreciate it if you could provide your final scores and any additional comments before the rebuttal period concludes**. Many thanks.
> > > >
> > > > Best regards
> > > >
> > > > Authors

---

### Official Review · Reviewer_ZqfG · 2024-11-05

**Soundness:** 3
**Presentation:** 2
**Contribution:** 2
**Rating:** 5
**Confidence:** 3

**Summary:**

This paper proposes a white-box neural network watermarking method NeuralMark, which aims to protect the network from three typical attacks. To offer resistance to forging attacks, NeuralMark establishes a hash-based mapping between keys and watermarks. It selectively embeds watermarks into model parameters to counter overwriting attacks and uses average pooling to defend against fine-tuning and pruning. The security and effectiveness of the method have been empirically verified on multiple architectures and various tasks.

**Strengths:**

- Comprehensive experiments demonstrate the scalability of the proposed NeuralMark across 14 models, covering five image tasks and one text generation task.
- The paper provides a theoretical analysis to determine the security boundary.

**Weaknesses:**

- The paper lacks consideration for the adaptive attack, which are crucial for robustness. A naïve adaptive attack can freeze model parameters and minimize the $L_e$ loss to learn the key and watermark, potentially compromising model ownership.
- The paper lacks a detailed description of attack methods, particularly whether fine-tuning and pruning are applied to all parameters or only the Watermark Filtering layer. This could affect the resilience of the method.
- The resistance to overwriting attacks for NeuralMark is unclear. In Section 5.3, the “Overwriting Attack” paragraph states that “the adversary’s watermark detection rate reaches 100%”, which, even if matched by the original watermark, still leaves model ownership ambiguous.
- Limitations of the proposed method are not explicitly discussed.
- Experimental comparisons with other methods are insufficient. For instance, in Figure 3, only resistance to pruning attacks at various ratios is shown for NeuralMark without comparisons to other methods. Additionally, Table 5 only compares against VanillaMark, lacking comparisons with GreedyMark.

**Questions:**

- Can the proposed method defend against adaptive attacks? Additional experiments to assess its resilience to adaptive attacks are necessary.
- How are the three types of attacks applied? Are fine-tuning and pruning applied to all parameters, or only to the Watermark Filtering layer? If the latter, would the method still be effective against attacks?
- Can this method resist overwriting attacks? With both the original and adversarial watermark detection rates at 100%, there is ambiguity in model ownership.
- What are the limitations of NeuralMark?
- Can the experimental comparisons with other methods be expanded? For example, comparative experiments with other methods need to be added in Figure 3 and Table 5.

---

> ### Author Response · Authors · 2024-11-24
> **Rebuttal-Part I**
>
> Thank you for your thoughtful review and valuable feedback. We address your concerns as follows.
>
> > W1. The paper lacks consideration for the adaptive attack, which are crucial for robustness. A naïve adaptive attack can freeze model parameters and minimize the loss to learn the key and watermark, potentially compromising model ownership. Can the proposed method defend against adaptive attacks? Additional experiments to assess its resilience to adaptive attacks are necessary.
>
> **AW1:** We appreciate your insightful suggestion. Indeed, we have considered the **adaptive attack**, which **we refer to reverse engineering attack in our manuscript** (please see the details in Section 3.3 in the manuscript). Such attacks involve first randomly forging a counterfeit watermark and then deriving a corresponding secret key by freezing the model parameters.
> **To defend against such attacks, we establish a hash mapping relationship between the secret key and the watermark**.
>
> On the one hand, if an adversary attempts to forge a pair of counterfeit secret key and watermark through reverse engineering **while considering the hash mapping relationship**, it is **computationally infeasible** due to the **avalanche effect** of hash functions, where even small changes in the input result in significantly different outputs [1]. As a result, **any attempt to learn the secret key and watermark would require breaking the underlying cryptographic hash function, as detailed in Strength 1 from Reviewer 33YR**.
>
> On the other hand, if an adversary forges a pair of counterfeit secret key and watermark through reverse engineering **without considering the hash mapping relationship**, the adversary may achieve a watermark detection rate exceeding the security threshold $\rho\^\ast$ **but will fail to satisfy the hash mapping relationship**. However, the legitimate model owner can present a valid pair of secret key and watermark that **not only exceeds $\rho^\ast$, but also satisfies the hash mapping relationship**. As established in Theorem 1, the probability of such an occurrence occurring by chance is negligible, **providing strong cryptographic evidence to support third-party verification agencies in correctly determining the model's ownership**. Consequently, we have defined the following conditions for watermark verification in NeuralMark (please see the details in Watermark Verification in `Section 4.2`):
> >(i) The watermark detection rate exceeds a theoretical security boundary, which will be analyzed later; and (ii) The watermark must correspond to the output of the hash function applied to the secret key, ensuring cryptographic consistency with the predefined hash mapping.
>
> To further clarify this mechanism, we have summarized the watermark verification process in `Algorithm 2` within `Appendix A` in the revision. In summary, **the forging attack through reverse engineering in NeuralMark is infeasible, regardless of whether the hash mapping relationship is considered**. Therefore, to evaluate its resilience against forging attacks, we have used 10 sets of randomly forged watermarks to directly verify them with the watermarked model in the experiments. Table 1 shows the watermark detection rates of forging attacks, we can see that NeuralMark demonstrates robust resistance against forging attacks (Please see the details in `Forging Attack in Section 5.3` of the revision). Additionally, we have revised the relevant statements in `lines 178–185` and `Section 3.3`, and included `Appendix B.2` in the revision.
>
> **Table 1. Comparison of resistance to forging attacks using ResNet-18**
> | Dataset         | NeuralMark | VanillaMark | GreedyMark | VoteMark |
> |-----------------|------------|-------------|------------|----------|
> | CIFAR-10        | 48.56      | 100.00      | 50.70      | 100.00   |
> | CIFAR-100       | 49.41      | 100.00      | 52.85      | 100.00   |
>
> [1] Hanwen Liu, Zhenyu Weng, Yuesheng Zhu, and Yadong Mu. 2023. Trapdoor normalization with irreversible ownership verification. ICML’23.

---

> ### Author Response · Authors · 2024-11-24
> **Rebuttal-Part II**
>
> > W2. The paper lacks a detailed description of attack methods, particularly whether fine-tuning and pruning are applied to all parameters or only the Watermark Filtering layer. This could affect the resilience of the method. How are the three types of attacks applied? Are fine-tuning and pruning applied to all parameters, or only to the Watermark Filtering layer? If the latter, would the method still be effective against attacks?
>
> **AW2:** Thank you for your valuable feedback. **For all fine-tuning attacks, we follow [1] and use the same hyper-parameters as during training, except for setting the learning rate to 0.001. In the manuscript, we replace the task-specific classifier and minimize the main task loss $\mathcal{L}\_m$ to optimize all parameters for 100 epochs.** In addition, as you suggested, we have conducted experiments where only the watermark embedding layer is fine-tuned, and the specific results are shown in the Table 2.
>
> **Table 2. Comparison of resistance to fine-tuning attacks against watermark embedding layer using ResNet-18. Values (\%) inside and outside the bracket are watermark detection rate and classification accuracy, respectively.**
> | Fine-tuning              | Clean (AlexNet/ResNet-18) | NeuralMark (AlexNet/ResNet-18) | VanillaMark (AlexNet/ResNet-18) | GreedyMark (AlexNet/ResNet-18) | VoteMark (AlexNet/ResNet-18) |
> |--------------------------|--------------------------|--------------------------------|--------------------------------|-------------------------------|-------------------------------|
> | CIFAR-100 to CIFAR-10    | 85.55 / 89.15           | 85.35 (100) / 88.83 (100)     | 85.48 (91.01) / 89.35 (85.93)  | 80.41 (96.48) / 76.15 (94.14) | 84.97 (89.06) / 89.66 (85.54) |
> | CIFAR-10 to CIFAR-100    | 58.96 / 49.74           | 58.50 (100) / 49.77 (100)     | 58.75 (74.21) / 49.97 (70.31)  | 51.75 (97.65) / 19.94 (82.42) | 58.81 (80.07) / 49.08 (71.87) |
> | Caltech-256 to Caltech-101 | 47.65 / 74.09         | 71.29 (100) / 73.12 (100)     | 71.56 (100) / 74.03 (100)      | 72.04 (100) / 68.45 (100)     | 71.62 (100) / 72.47 (99.60)   |
> | Caltech-101 to Caltech-256 | 40.61 / 40.00         | 40.34 (100) / 40.34 (100)     | 40.71 (96.09) / 39.04 (93.36)  | 40.68 (100) / 36.45 (98.82)   | 39.52 (95.31) / 39.73 (93.75) |
>
> As can be seen, the watermark detection rate remains at 100\%, but the model performance exhibits a substantial decline. Specifically, for the CIFAR-10 to CIFAR-100 task using ResNet-18, the accuracy achieved by NeuralMark through fine-tuning the watermark embedding layer is 49.77\%, which is markedly lower than the 71.67\% accuracy obtained when all parameters are fine-tuned. Similar trends are observed across other methods. Those results indicate that solely fine-tuning the watermark embedding layer and classifier makes it challenging to ensure effective model performance. Consequently, we do not consider this type of fine-tuning attack in the subsequent experiments.
>
> **For pruning attacks, we randomly reset a specified proportion of model parameters in the watermark embedding layer to zero**. We have added the above experimental details and results in the revision, please refer to `lines 372-375` and `406-408`, and `Appendix E.1` for details.
>
> [1] Liu, H., Weng, Z., Zhu, Y. (2021). Watermarking Deep Neural Networks with Greedy Residuals. ICML’21.

---

> ### Author Response · Authors · 2024-11-24
> **Rebuttal-Part III**
>
> > W3. The resistance to overwriting attacks for NeuralMark is unclear. In Section 5.3, the “Overwriting Attack” paragraph states that “the adversary’s watermark detection rate reaches 100\%”, which, even if matched by the original watermark, still leaves model ownership ambiguous. Can this method resist overwriting attacks? With both the original and adversarial watermark detection rates at 100\%, there is ambiguity in model ownership.
>
> **AW3:** We are sorry for this confusion. For overwriting attacks, if an adversary only embeds a counterfeit watermark through modifications to the model parameters without removing the original one, the resulting model will contain both the original and counterfeit watermarks. In this case, **the model owner can submit a model containing only the original watermark** to an authoritative third-party verification agency. In contrast, **the adversary cannot provide a model with only the counterfeit watermark, as they have not successfully removed the original watermark**. Accordingly, **the adversary cannot prove innocence unless they develop a new model embedded with only their counterfeit watermark**. This not only makes stealing the original model unnecessary but also incurs significant training costs.
>
> In addition, this insight is supported by the consensus in the literature [1]. Specifically, on page 4 of 12 in [1]:
>
> > Some researchers believe embedding new watermarks as an effective white-box attack. They argue that once multiple watermarks exist in the network, the copyright should be shared by all the owners of those watermarks. In fact, this is not reasonable in the case of an authoritative third-party certifier existing. Because one of the watermark embedders can provide the trusted third party with a network containing only one authenticated watermark to prove that he or she owns the copyright exclusively. Therefore, we believe that simply embedding a new watermark is not enough as a successful attack.
>
> In summary, **we believe that for an overwriting attack involving modifications to model parameters to succeed, the original watermark must be removed. Otherwise, the overwriting attack fails, which is defined as the attack success Level III in our manuscript**. We have clarified the above statements in the revision, please refer to `Section 3.2` for details.
>
> [1] Renjie Zhu, Xinpeng Zhang, Mengte Shi, and Zhenjun Tang. Secure neural network watermarking protocol against forging attack. EURASIP Journal on Image and Video Processing, 2020:1–12, 2020.
>
> > W4. Limitations of the proposed method are not explicitly discussed. What are the limitations of NeuralMark?
>
> **AW4:** Thanks for your insightful suggestion. We have discussed the limitations and border impact of NeuralMark in `Appendix B.4` in the revision. The specific details are as follows.
>
> **Limitations and Broader Impact**
>
> Although NeuralMark demonstrates promising results and can be seamlessly integrated into various architectures, it has certain limitations. Specifically, it requires direct access to the model parameters, making it unsuitable for verifying ownership through a remote Application Programming Interface (API) where model parameters remain inaccessible.
> To address this limitation, a potential solution involves integrating NeuralMark with black-box NNW watermarking methods, such as those proposed in [1–2]. Specifically, trigger samples can be utilized alongside vanilla training samples to train the model while embedding the watermark through NeuralMark. This method enables the initial verification of model ownership by evaluating the prediction performance of trigger samples via the remote API. Based on this preliminary evidence, a formal request can be made to the API service provider for access to the corresponding model parameters. Once obtained, NeuralMark can be employed for a secondary, white-box verification to conclusively confirm model ownership. The practical implementation of this combined method is beyond the scope of this work and will be explored in future research.
>
> Ownership protection of artificial intelligence models is a critical and pressing issue. This paper presents a simple yet general method to safeguard model ownership. Our work aims to inspire further academic research in this vital area and advance industry adoption to effectively address ownership concerns related to models.
>
> [1] Lixin Fan, Kam Woh Ng, and Chee Seng Chan. Rethinking deep neural network ownership verification: Embedding passports to defeat ambiguity attacks. In NeurIPS, volume 32, 2019.
>
> [2] Lixin Fan, Kam Woh Ng, Chee Seng Chan, and Qiang Yang. Deepipr: Deep neural network ownership verification with passports. IEEE Transactions on Pattern Analysis and Machine Intelligence,
> 44 (10):6122–6139, 2021.

---

> ### Author Response · Authors · 2024-11-24
> **Rebuttal-Part IV**
>
> > W5. Experimental comparisons with other methods are insufficient. For instance, in Figure 3, only resistance to pruning attacks at various ratios is shown for NeuralMark without comparisons to other methods. Additionally, Table 5 only compares against VanillaMark, lacking comparisons with GreedyMark. Can the experimental comparisons with other methods be expanded? For example, comparative experiments with other methods need to be added in Figure 3 and Table 5.
>
> **AW5:** Thanks for your valuable comment. Since GreedyMark does not need the secret key for verification that can effectively resist forging attacks, we did not include GreedyMark for comparison in our manuscript. For completeness, **we have incorporated GreedyMark and VanillaMark for comparison as you suggested. Furthermore, we have added a recent method, VoteMark [1], as suggested by Reviewer 33YR**. VoteMark introduces random noises to embed the watermark and then employs a majority voting scheme to aggregate the results of multiple rounds of verification. However, it remains ineffective in resisting forging and overwriting attacks. Several key experimental results are reported as follows.
>
> **Table 3. Comparison of resistance to forging attacks after fine-tuning attacks and pruning attacks
> (with a pruning ratio of 40%) using ResNet-18.**
> | Dataset   | NeuralMark (Fine-tuning + Forging / Pruning + Forging) | VanillaMark (Fine-tuning + Forging / Pruning + Forging) | GreedyMark (Fine-tuning + Forging / Pruning + Forging) | VoteMark (Fine-tuning + Forging / Pruning + Forging) |
> |-----------|-------------------------------------------------------|--------------------------------------------------------|-------------------------------------------------------|------------------------------------------------------|
> | CIFAR-10  | 48.90 / 49.14                                         | 100.00 / 100.00                                        | 49.30 / 49.30                                         | 100.00 / 100.00                                      |
> | CIFAR-100 | 48.82 / 49.37                                         | 100.00 / 100.00                                        | 49.30 / 50.27                                         | 100.00 / 100.00                                      |
>
> **Table 4. Comparison of resistance to pruning attacks at various pruning ratios on the CIFAR-100 dataset using AlexNet and ResNet-18, respectively.**
> | Pruning Rate | NeuralMark          | VanillaMark         | GreedyMark          | VoteMark            |
> |--------------|---------------------|---------------------|---------------------|---------------------|
> | 10%     | 75.98 (100)          | 76.06 (100)          | 74.6 (98.43)         | 76.28 (100)          |
> | 30%     | 72.18 (99.61)        | 73.76 (99.21)        | 59.79 (92.18)        | 74.82 (100)          |
> | 50%     | 61.41 (99.61)        | 65.23 (99.21)        | 49.9 (80.46)         | 67.46 (100)          |
> | 70%     | 30.61 (99.61)        | 40.02 (94.53)        | 18.51 (73.04)        | 46.26 (97.65)        |
> | 90%     | 4.09 (98.82)         | 7.5 (79.29)          | 2.14 (60.15)         | 6.52 (80.46)         |
>
> From the results, we can see that NeuralMark can effectively resist forging and pruning attacks in all scenarios compared to baseline methods. Please refer to `Tables 4, 5, 6, and 7` and `Figures 3, 6, 7, and 8` in the revision for more detailed experimental results.
>
> [1] Fangqi Li, Haodong Zhao, Wei Du, and Shilin Wang. Revisiting the information capacity of neural network watermarks: Upper bound estimation and beyond. In AAAI’24.

---

> > ### Author Response · Authors · 2024-11-30
> > **Request for Reconsideration of the score**
> >
> > Dear Reviewer ZqfG
> >
> > We sincerely appreciate the insightful and constructive feedback you have provided. We have made a thorough and diligent effort to address all of your concerns, and we believe these revisions have significantly enhanced the quality of our manuscript. **We are deeply grateful for your valuable suggestions, and if, after reviewing the revised manuscript, you feel it now meets your expectations, we would be sincerely thankful if you could consider raising your score**.
> >
> > Should you have any further questions or require additional clarifications, please do not hesitate to reach out, and we would be happy to assist before the rebuttal deadline.
> >
> > Best regards
> >
> > Authors

---

> > > ### Author Response · Authors · 2024-12-03
> > > **Request for Final Feedback**
> > >
> > > Dear Reviewer ZqfG:
> > >
> > > Thank you once again for your thoughtful and constructive review of our manuscript. We sincerely hope that all reviewers will approach the entire review process with the utmost responsibility. **We would greatly appreciate it if you could provide your final scores and any additional comments before the rebuttal period concludes**. Many thanks.
> > >
> > > Best regards
> > >
> > > Authors

---

### Author Response · Authors · 2024-11-24
**Rebuttal Summary**

Dear Reviewers:

We sincerely thank all reviewers for their time and effort in reviewing our work and for providing constructive and valuable comments to enhance the quality of the paper.

First, we are pleased to observe that the reviewers find that the following aspects:

*  Hash mapping ties the secret key directly to the watermark to make reverse engineering infeasible, as altering the watermark would require breaking the hash (**Reviewer 33YR**). Watermark filtering limits parameter overlap, making overwriting attacks minimize interference with the original watermark (**Reviewer 33YR**). Using a watermark as a filter for injected parameters makes sense (**Reviewer 2mAv**).

*  This paper includes a theoretical analysis of the security boundaries for NeuralMark (**Reviewers ZqfG and 2mAv**), extends model watermarking to the Transformer architectures (**Reviewer 464P**), and provides extensive experimental results (**Reviewers 33YR, ZqfG, and 2mAv**).

*  The research direction about model watermarks is important, and the presentation is clear and well-written (**Reviewer 464P**).

Then, we have addressed all concerns raised by all reviewers and conducted additional experiments in the revision in response to your valuable suggestions:

*  To **Reviewer ZqfG**: We have conducted the adaptive attack, *i.e.*, reverse engineering (`Forging Attack in Section 5.3`), performed fine-tuning experiments with the watermark embedding layer (`Appendix E.1`), explained the success criteria for overwriting attacks (`Section 3.2`), discussed the limitations and broader impact (`Appendix B.4`), and included experimental results for the baseline methods in resisting attacks (`Tables 4, 5, 6, 7 and Figures 3, 6, 7, 8`).

*  To **Reviewer 33YR**: We have clarified the motivation behind our interest in the weight-based watermarking field (`Lines 048-053 and 108-112`), incorporated a recent baseline method [1], performed experiments to evaluate the robustness of watermark filtering with distinct filtering rounds (`Appendix E.6`), and detailed the fine-tuning attacks (`Removal + Forging Attack in Section 5.3`).

*  To **Reviewer 464P**: We have highlighted the contributions of NeuralMark (`Lines 069-074`), analyzed how to support repeated public verification in NeuralMark (`Lines 211-212 and Appendix B.1`), and provided the source codes and necessary documents at  https://anonymous.4open.science/r/NeuralMark.

*  To **Reviewer 2mAv**: We have explained the function of the key-to-watermark hash mapping (`Lines 178-185 and Appendix B.2`), highlighted the contributions of NeuralMark (`Lines 069-074`), and detailed the robustness of NeuralMark against forging attacks under various scenarios (`Forging Attack in Section 5.3 and Appendix E.3`).

In summary, we believe that those revisions and clarifications have strengthened our manuscript, and we hope that our responses meet your satisfaction. Please find our detailed replies to your specific points below, along with the revised version of our manuscript. Thanks again.

[1] Fangqi Li, Haodong Zhao, Wei Du, and Shilin Wang. Revisiting the information capacity of neural network watermarks: Upper bound estimation and beyond. In AAAI'24.

Best regards,

Authors

---

### Author Response · Authors · 2024-11-29
**Looking forward to your feedback on our revisions**

Dear Reviewers:

We sincerely appreciate your valuable feedback and insightful comments, which have greatly contributed to improving our work.

**We fully understand that the review process can be time-consuming, and we are truly grateful for the time and effort you've invested**.
As some time has passed since we submitted our revised version, we would like to kindly follow up to inquire if there are any additional questions, concerns, or clarifications needed from our side to assist in the review process.

Please feel free to reach out to us at your convenience, and we are more than happy to provide any further information if necessary. Many thanks!

Best regards,

Authors

---

### Author Response · Authors · 2024-12-04
**Final Request from Authors**

Dear Reviewers, ACs, SACs, PCs:

First, we sincerely thank all the reviewers for their valuable feedback. During the rebuttal period, **although we did not receive direct responses, we carefully addressed each concern raised in the initial reviews and believe we have fully addressed all issues**. Additionally, as reviewers, we have taken our responsibilities seriously, diligently fulfilling our duties and remaining committed to upholding the integrity of the review process. Finally, **given the borderline status of our work, we respectfully request that it be fairly evaluated during the final decision**.

Best regards,

Authors

---

### Note · Authors · 2025-01-30

I have read and agree with the venue's withdrawal policy on behalf of myself and my co-authors.